# More Bang for the Buck: Process Reward Modeling with Entropy-Driven Uncertainty

## Abstract

We introduce the Entropy-Driven Uncertainty Process Reward Model (EDU-PRM), a novel entropy-driven training framework for process reward modeling that enables dynamic, uncertainty-aligned segmentation of complex reasoning steps, eliminating the need for costly manual step annotations. Unlike previous Process Reward Models (PRMs) that rely on static partitioning and human labeling, EDU-PRM automatically anchors step boundaries at tokens with high predictive entropy, effectively capturing intrinsic logical transitions and facilitating efficient exploration of diverse reasoning paths. On the ProcessBench benchmark, EDU-PRM outperforms strong public PRM baselines, such as Math-Shepherd PRM and Omega PRM, and EDU-PRM achieves comparable results with SOTA models while only using $1.5\%$ training data. Furthermore, by leveraging our proposed EDU sampling strategy, we observe accuracy boosts from $64.7\%$ to $67.3\%$ for generative reasoning tasks, accompanied by a reduction of $32\%$ in token usage. These findings underscore the potential of EDU-PRM as a scalable and annotation-efficient paradigm for process supervision in mathematical reasoning, paving the way for more efficient and robust approaches to complex mathematical problem solving.

## 1 Introduction

Large Language Models (LLMs), such as GPT-4o (OpenAI et al., 2024) and Deepseek-V3 (DeepSeek-AI et al., 2024), have achieved remarkable performance across a wide range of tasks, particularly in natural language understanding and generation. Despite these successes, LLMs still struggle with complex multi-step reasoning problems, where verifying each intermediate reasoning step is essential to producing reliable solutions (Wei et al., 2022). To address these challenges, recent approaches adopted reinforcement learning (RL) (Murphy, 2024) with reward models, moving from supervision focused solely on final answers to more granular and step-level evaluations using LLM judges.

Process Reward Models (PRMs) (Lightman et al., 2024) represent a significant step forward by providing stepwise feedbacks, improving both the reliability and the interpretability for model reasoning. However, the deployment of PRMs introduces two critical challenges. **First**, obtaining high-quality step-level data is difficult: defining what constitutes a "correct" intermediate step is often ambiguous, and large-scale human annotation, as used in datasets like PRM800K (Lightman et al., 2024), is time-consuming and costly. Though recent methods, such as Qwen2.5-PRM (Zheng et al., 2025; 2023), employ LLM-based judgment or Monte Carlo estimation (Xie et al., 2024; Zhang et al., 2024) to scale supervision, these approaches still demand substantial computational resources. **Second**, the reliability of intermediate evaluation remains limited: PRMs can be "cheating", as high step scores do not always guarantee a correct final answer (DeepSeek-AI et al., 2024). This undermines the effectiveness of stepwise supervision and poses a significant barrier to robust reasoning.

To overcome these challenges, we propose **Entropy-Driven Uncertainty Process Reward Model (EDU-PRM)**, a novel framework for scalable and efficient step-level supervision without the need for expensive human or LLM annotation. Our approach leverages entropy-driven sampling to automatically generate diverse and informative intermediate steps, addressing the data scarcity problem. Furthermore, by explicitly modeling uncertainty, EDU-PRM improves the alignment between stepwise evaluation and final answer correctness, thereby mitigating the "cheating" issue.

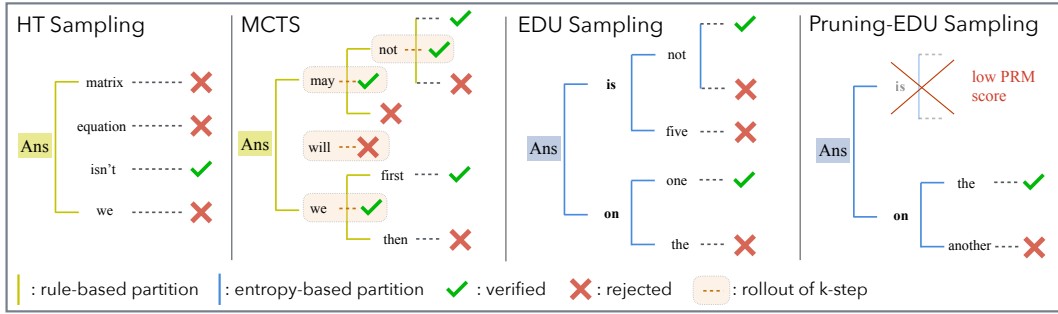

Figure 1: Comparison of sampling methods in Process Reward Models (PRMs). **High Temperature (HT) sampling** performs exhaustive sampling and selects the best answer from $N$ candidates (Best-of-N), yet incurs substantial computational overhead $\mathcal{O}(N)$ and and risks overlooking high-quality solutions due to random sampling. **OmegaPRM** mitigates this by integrating Monte Carlo Tree Search (MCTS) for localized trajectory assessment and pruning, thereby reducing search complexity. However, these sampling methods rely on rule-based partitioning and random initial candidate generation. **Entropy-Driven Uncertainty (EDU) Sampling** strategically generates candidates via high-entropy words (e.g., "is", "on"), thereby achieving reduced complexity $\mathcal{O}(\log(N))$ and enabling a more deterministic exploration of reasoning paths. **Pruning-EDU Sampling**, incorporates targeted pruning mechanisms to minimize "cheating" vulnerabilities—such as premature convergence on low-PRM-score trajectories—while further optimizing token efficiency for EDU.

Our main contributions are as follows.

**EDU Sampling for PRM Training:** We propose an entropy-driven uncertainty (EDU) sampling strategy to automatically generate diverse and informative step-level data, which is directly used to train Process Reward Models. This approach eliminates the need for costly human or LLM annotation and enables scalable and high-quality supervision.

**Reliable Stepwise Supervision:** PRMs trained with EDU sampling achieve substantially better alignment between stepwise evaluation and final answer correctness, effectively mitigating the "cheating" issue and enhancing the reliability of step-level supervision.

**Efficient and Accurate Solution Generation:** Applying EDU sampling during inference leads to higher accuracy and lower token consumption compared to conventional high-temperature sampling methods.

In summary, EDU-PRM enables scalable, annotation-efficient, and reliable step-level supervision for complex reasoning tasks.

## 2 RELATED WORKS

Methods for evaluating LLM outputs have evolved from early rule-based heuristics to sophisticated model-based reward frameworks. Initial approaches (Mu et al., 2024) relied on keyword matching, which limited their generalizability when domain transferring. The LLM-as-judge paradigm (Zheng et al., 2023) enabled self-evaluation but introduced self-verification biases, as well as increased computational costs (Wang et al., 2023).

Output-Reward Models (ORMs; Wang et al. (2024a); Yuan et al. (2024); Luo et al. (2024b)) assign scores to final outputs based on human annotation. However, ORMs often neglect intermediate reasoning steps, risking misjudgment when flawed processes yield correct results. To address this, Process Reward Models (Lightman et al., 2024; Zhang et al., 2025) score reasoning chains at the sub-step level, using either soft labels (LLM-generated scores) or hard labels (expert binary judgments). Soft labels enable scalable annotation but may introduce bias, while hard labels offer reliability at a higher cost. PRMs improve reliability in tasks such as mathematical reasoning by penalizing erroneous intermediate steps.

Despite progress, key challenges remain, including the difficulty of obtaining high-quality labels and the limited effectiveness of current PRM approaches (DeepSeek-AI et al., 2025; Wu et al., 2024; Sun et al., 2024; Yin et al., 2025). Addressing these issues has inspired diverse PRM architectures.

Math-Shepherd PRM (Wang et al., 2024c) employs a two-stage process: the base model generates solution traces via self-consistency sampling, and a symbolic checker verifies answers and propagates binary labels to intermediate steps. This automatic chain annotation reduces manual effort and supports efficient PRM training.

Omega PRM (Luo et al., 2024a) frames problem-solving procedure as a search tree, using Monte-Carlo Tree Search to decompose tasks and explore promising branches. PRM predictions guide tree exploration and serve as rewards during policy optimisation, enhancing exploration efficiency and reasoning capability.

## 3 METHOD

### 3.1 MOTIVATION AND OVERVIEW

As discussed in Section 2, existing PRMs have made substantial progress but still face several critical challenges, such as the difficulty of obtaining high-quality labels and the limited effectiveness of predicting final answers. In particular, many conventional PRMs rely on superficial textual cues such as blank lines or punctuation to segment reasoning steps and to assign rewards. However, these heuristics fail to capture the underlying logical transitions in complex solution traces, resulting in suboptimal supervision and limited generalization.

Recent advances in reasoning with LLMs have highlighted the importance of stepwise exploration during solution generation. In particular, Chain-of-Thought (CoT) Decoding (Wang & Zhou, 2024) demonstrates that branching at token positions where the model exhibits uncertainty, specifically the probability gap between the top-1 and top-2 candidates is small, can reveal alternative reasoning paths and improve overall solution quality. Building on this insight, studies such as Cheng et al. (2025) further establish that high-entropy tokens serve as natural anchors for meaningful exploration. These tokens often correspond to logical pivots or transitions in the reasoning process, making them ideal candidates for step segmentation and branching.

Motivated by these findings, we propose placing token-level entropy at the core of our segmentation and sampling strategy to build PRMs. By dynamically identifying and branching at positions of maximal uncertainty, our Entropy-Driven Uncertainty Process Reward Model (EDU-PRM) is able to generate logically coherent, diverse, and informative step-level data. This approach not only enhances the quality of process supervision but also reduces reliance on manual annotation and rigid heuristics, paving the way for more robust and scalable reward modeling.

Furthermore, although soft labels may introduce more noise compared to hard labels, Omega PRM (Luo et al., 2024a) has empirically demonstrated that using soft labels achieves a significantly higher accuracy ($70.1\%$) than hard labels ($63.3\%$) in process supervision accuracy. Therefore, despite the potential for increased noise, all of our experiments consistently adopt soft labels for step-level reward assignment in this paper.

### 3.2 ENTROPY-DRIVEN UNCERTAINTY SAMPLING

Token-level entropy quantifies the model's uncertainty in predicting the next token at each decoding step. High entropy indicates that the model's probability distribution over possible next tokens is more dispersed, reflecting greater ambiguity or indecision. In contrast, low entropy suggests the model is confident, with most probability mass assigned to a single token.

During the reasoning process, increased entropy often signals points where the model is less certain about how to proceed. EDU sampling leverages these high-entropy tokens as *uncertainty anchors*, guiding the segmentation of reasoning steps to better reflect the underlying logical structure of the solution trace, rather than relying on superficial textual cues.

Formally, we apply the softmax function to the output logits of an autoregressive model at each decoding step, yielding a probability distribution $P_v$ over possible next tokens $v$ (Kwon et al., 2023;

Aminabadi et al., 2022). Then, the entropy at the next position $v$ is calculated as:

$$H_v = -\sum_v P_v \cdot \log\left(P_v + \epsilon\right) \tag{1}$$

where $\epsilon$ is a small constant for numerical stability.

We define position $v$ as an **uncertainty anchor** when $H^{(v)}$ exceeds an adaptive threshold $\tau(\mathbf{H})$, which is dynamically adjusted according to the maximum sampling branch number in the sampling process (see Section 5 for further analysis).

Overall, as illustrated in Figure 1, our EDU sampling workflow consists of two main stages: 1) entropy-based anchor detection and branching, and 2) fragment-level evaluation and labeling.

**EDU Sampling at Anchor Position**   To balance solution diversity and quality, EDU sampling repeats branching an anchor position of only top-2 logits at the first token and each anchor position afterwards,[1] with subsequent tokens generated greedily (i.e. $\arg\max_t \mathbf{P}_v^{(t)}$) until the next anchor position is reached. This strategy efficiently samples alternative reasoning paths without excessive computational overhead. To avoid artifacts caused by mathematical symbols (e.g., $\sum, \int$), we exclude tokens in the symbol set $\mathcal{S}$ (see Appendix A.4) from entropy calculations. In our experiments, we observed that branching at these tokens often leads to garbled outputs.

**Monte Carlo Estimation Scoring**   After performing the EDU sampling, each answer is segmented into multiple fragments at anchor positions. For each fragment, we assign a correctness label ([0, 1]) based on the final solution's validity using Monte Carlo Estimation (MCE; (Katzgraber, 2011)). This fragment-level approach enables a fine-grained assessment of reasoning steps, as shown in Figure 1, where each segment is mapped to its corresponding correctness label.

### 3.3 ENTROPY-DRIVEN UNCERTAINTY PRM

Consequently, we can perform the EDU sampling workflow to construct the EDU-PRM training dataset, where each instance consists of a triple: a question, a solution or a solution fragment, and an associated label indicating the correctness of the solution. This structure allows the model to learn not only from complete solutions but also from partial reasoning steps, thereby enhancing its ability to generalise and identify robust reasoning patterns.[2] We then train EDU-PRM via a classification-oriented cross-entropy loss, $\mathcal{L} = -\frac{1}{N}\sum_{i=1}^{N}\sum_{k=0}^{1} y_{ik} \log p_{ik}$, where $N$ is the number of examples, $y_{ik}$ are the target labels, and $p_{ik} = \mathrm{softmax}(\mathbf{z}_i)_k$ denotes the predicted probabilities from logits $\mathbf{z}_i$. This framework enables EDU-PRM to learn to discriminate between correct and incorrect reasoning steps effectively.

## 4 EXPERIMENTS

In this section, we report the experimental results of the proposed EDU-PRM. In general, we perform two evaluation setups, a direct accuracy evaluation over PRM benchmarks and applying PRMs as a BoN results selector over a series of math reasoning tasks. In addition, we also experiment with the proposed EDU sampling strategy, comparing with the traditional high-temperature (HT) sampling method, focusing not only on accuracy but also on token efficiency, offering a more nuanced perspective beyond traditional metrics.

### 4.1 IMPLEMENTATIONS OF EDU-PRM

We first describe the implementation and training details of the proposed EDU-PRM, as well as the compared methods. Our EDU-PRM implementation follows the methodology established in Math-Shepherd PRM (Wang et al., 2024c) and Omega PRM (Luo et al., 2024a), with consistent experimental settings and parameter configurations.

---

[1]Experiments with top-3 and other schemes yielded similar results.

[2]For the sake of clarity and brevity, unless explicitly stated otherwise, all references to EDU-PRM or Greedy EDU-PRM in this paper refer to the specific method described in the Method section.

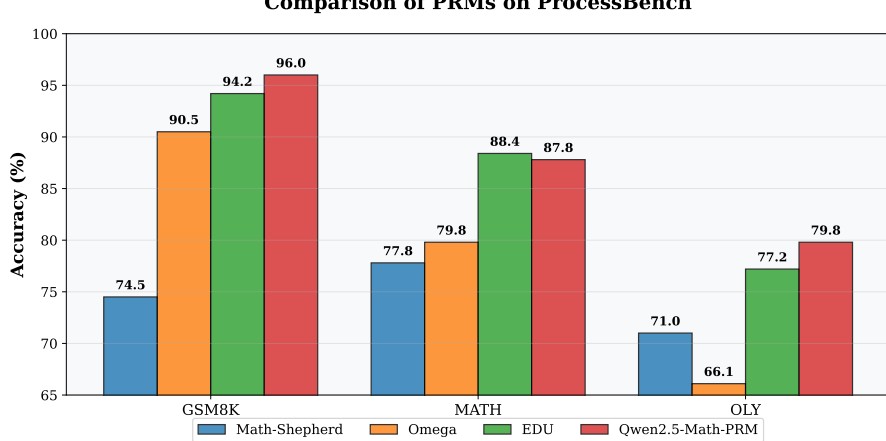

Figure 2: Accuracy comparison on ProcessBench for four 72B-parameter PRMs: Math-Shepherd PRM, Omega PRM, EDU PRM, and Qwen2.5-Math-PRM. As a competitive PRM method, our proposed EDU PRM attains the highest accuracy on the MATH test dataset. On GSM8K and OLY datasets, EDU PRM matches the performances of Qwen2.5-Math-PRM.

For detailed model training, we use data from the MATH training set (Hendrycks et al., 2021), selecting $7,500$ problems as the base query set and sampling up to $100$ candidate solutions per problem. After using the EDU sampling (token-level predictive entropy threshold $= 1.0$), the training dataset comprises approximately 1.42M instances, with a label distribution of $52\%$ hard and $48\%$ soft labels.

We train PRMs based on Qwen2.5-72B-Base and Qwen2.5-7B-Base (Qwen et al., 2025). All the details of the training frameworks, dataset statistics, and inference hyperparameters are listed in Appendix A.3, and the prompts used for solution verification are also provided in Appendix A.5.

### 4.2 Evaluation Benchmarks and Comparison Baselines

We evaluate the effectiveness of PRMs from two aspects, directly evaluating the accuracy of PRMs and a best-of-$N$ (BoN) selection via PRM scoring on real tasks. For accuracy evaluation, we utilise the ProcessBench (Zheng et al., 2025), containing questions, responses, and correctness labels, where PRMs aim to predict whether the response is correct or not. For the BoN selection evaluation, we choose several math reasoning benchmarks, including OlympiaBench (OLY) (He et al., 2024), MATH (Hendrycks et al., 2021), GSM8K (Cobbe et al., 2021), and CollegeMath (Tang et al., 2024). For each query, we generate $128$ candidate solutions using Qwen2-7B-Instruct (Yang et al., 2024a), and each response is scored by the PRMs, determining the best responses to the question.

We compare with sota PRMs, including Math-Shepherd-Mistral-7B-PRM (Wang et al., 2024b), Qwen2.5-Math-7B-PRM800K, Qwen2.5-Math-PRM-7B, Qwen2.5-Math-PRM-72B, and Qwen2.5-Math-RM-72B (Yang et al., 2024b). Note that the open-sourced versions of these baselines are trained on much larger datasets than ours. For fair comparison, we re-implement these baselines based on the same data and base models as EDU-PRM, except the Qwen2.5-Math-PRM series. We report the performance of the original version of Qwen2.5-Math-PRMs as strong sota baselines.

### 4.3 ProcessBench Evaluation of PRM Accuracy

Figure 2 demonstrates that EDU-PRM-72B achieves outstanding performance in solution correctness judgment across multiple benchmarks. On the MATH dataset, EDU-PRM-72B attains the highest judgment accuracy of $88.4\%$, outperforming Qwen-2.5-math-PRM-72B ($87.8\%$) by a margin of $0.6\%$. Additionally, EDU-PRM-72B exhibits robust judgment accuracy on GSM8K ($94.2\%$) and OlympicBench ($77.2\%$), further highlighting its effectiveness in verifying mathematical solutions. Notably, EDU-PRM-72B consistently surpasses Math-Shepherd PRM and Omega PRM across all evaluated benchmarks. Detailed experimental results are provided in Appendix A.2.

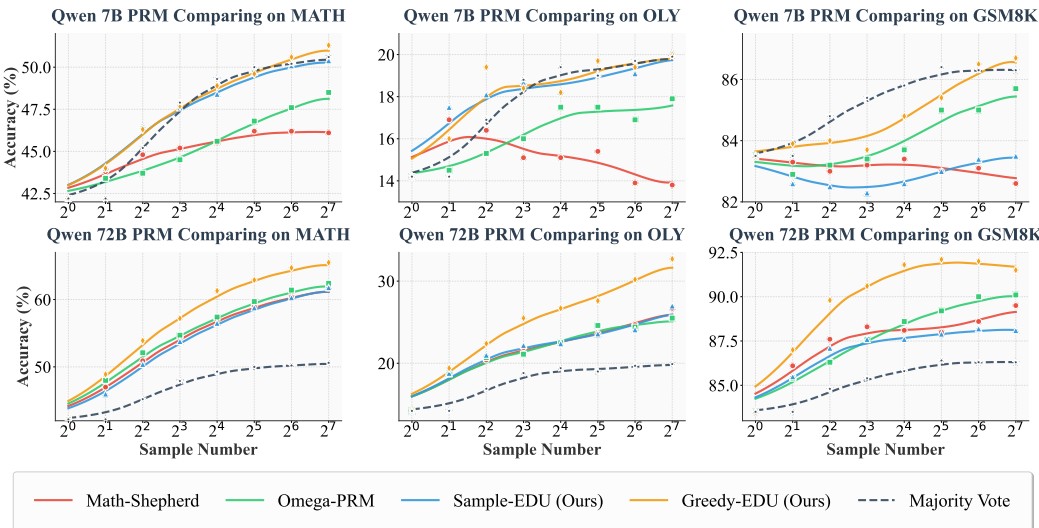

Figure 3: Comparison of PRM performance on the MATH, OLY, and GSM8K benchmarks for Qwen 7B and 72B models. Evaluated methods: Math-Shepherd, Omega-PRM, Sample-EDU, Greedy-EDU, Majority Vote serves as a non-PRM baseline. Markers show raw scores; curves are Gaussian-smoothed (trend visualisation only). **Greedy-EDU** consistently leads or matches the best results across datasets and model scales.

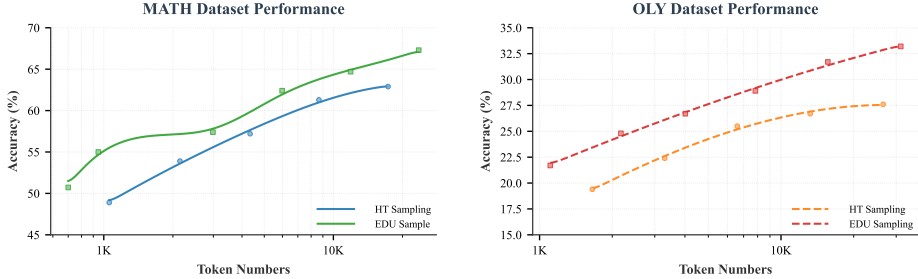

Figure 4: Comparison of sample strategies under the EDU-PRM 72B model on the MATH and OLY test sets: High-Temperature (HT) Sampling, EDU Sampling. Markers denote raw measurements; curves are Gaussian-smoothed trends. Points nearer the upper-left frontier indicate a better accuracy–token trade-off. It can be observed that on both the OLY and MATH test sets, EDU Sampling achieves an overall higher accuracy compared to HT Sampling while consuming fewer tokens.

## 4.4 EVALUATING PRMS VIA BON

Figure 3 summarises the performance of different models across three datasets, highlighting the superior results of Greedy-EDU PRM (i.e. EDU-7B and EDU-72B respectivly). We observed that EDU-72B achieves up to a 3.7% lead on MATH and a 5.7% lead on OLY consistently across different sampling sizes, compared with SOTA baselines. When compared with majority voting, usually considered as a strong baseline of BoN, our PRM-based method can consistently achieve better accuracy of response selection, especially when the model size increases. Full experimental results are detailed in Table 3.

## 4.5 SAMPLING STRATEGY COMPARISON: EDU SAMPING VS. HT SAMPLING

After establishing the superior performance of EDU-PRM, we further investigate different sampling strategies during the inference. Specifically, we compare proposed EDU sampling on its accuracy and token efficiency with the traditional HT Sampling (temperature = 0.7).

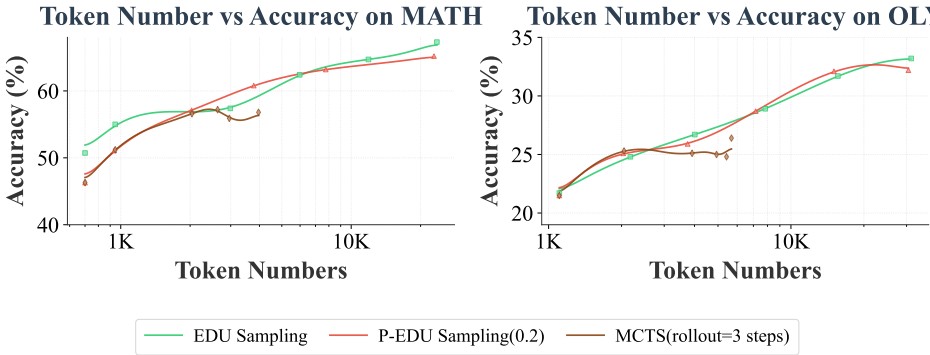

Figure 5: Comparison of sample strategies under the EDU-PRM 72B model on the MATH and OLY test sets: EDU Sampling, P-EDU Sampling (with a threshold of 0.2), and MCTS (with exploration depth not exceeding 3 steps rollout). Markers denote raw measurements; curves are Gaussian-smoothed trends. The x-axis represents token counts, and the y-axis represents accuracy (%). Points nearer the upper-left frontier indicate a better accuracy–token trade-off. P-EDU Sampling achieves a measurable lead on both the OLY and MATH test sets, yet EDU Sampling exhibits a more pronounced advantage under high token counts across both test sets.

Experimental results on the MATH and OLY test sets (see Figure 4) show that EDU sampling consistently outperforms HT sampling in both accuracy and token efficiency. On MATH, EDU sampling achieves 57.4% accuracy with 2,988 tokens, while HT sampling achieves 57.2% accuracy with 4,338 tokens on average. On OLY, EDU sampling attains 21.7% accuracy with 1,107 tokens, compared to 19.4% of HT sampling with 1,655 tokens.

Both methods initially show increasing accuracy with more tokens, however at higher token counts, EDU sampling maintains a steep upward trajectory in accuracy, while HT sampling improves plateaus, indicating diminishing returns. This highlights EDU sampling's superior capability to leverage additional tokens for sustained accuracy gains.

Overall, these results indicate that the EDU sampling not only achieves higher accuracy but also utilizes tokens more efficiently, making it a preferable strategy for mathematical reasoning tasks under computational constraints.

## 4.6 PRUNING-EDU SAMPLING VS MCTS WITH EDU SAMPLING

To further enhance solution generation efficiency, we introduce two advanced sampling strategies: Pruning-EDU (P-EDU) sampling, which applies a pruning threshold of 0.2 to filter out low-confidence branches, and Monte Carlo Tree Search (MCTS) with a rollout depth of 3 steps for strategic exploration. The motivation for pruning is that if the initial PRM score for a branch is very low, continued reasoning along this path is unlikely to yield correct solutions, so it is preferable to prune such branches early—provided at least one promising path is retained to ensure coverage. In contrast, MCTS leverages forward-looking exploration. By simulating future reasoning steps, it can make more informed decisions about which current paths are worth pursuing, rather than relying solely on immediate scores.

Table 6 and Figure 5 summarize the distinct performance profiles of these strategies on both the MATH and OLY test sets. EDU sampling's accuracy steadily increases with more tokens, while P-EDU sampling achieves a balanced trade-off between accuracy and token usage, reaching 32.1% accuracy at 15,050 tokens on OLY, comparable to EDU sampling in the mid-token range, benefited from the effective pruning of low-confidence paths. On MATH dataset, MCTS performs well in the low-token regime, achieving 51.2% accuracy at 946 tokens, similar to P-EDU sampling, which achieves 51.1% using 937 tokens on average.

Overall, these results demonstrate that the P-EDU sampling can outperform the standard EDU sampling, particularly when the PRM is able to accurately identify and prune low-confidence branches early in the reasoning process. Meanwhile, the performance ceiling of MCTS is inherently con-

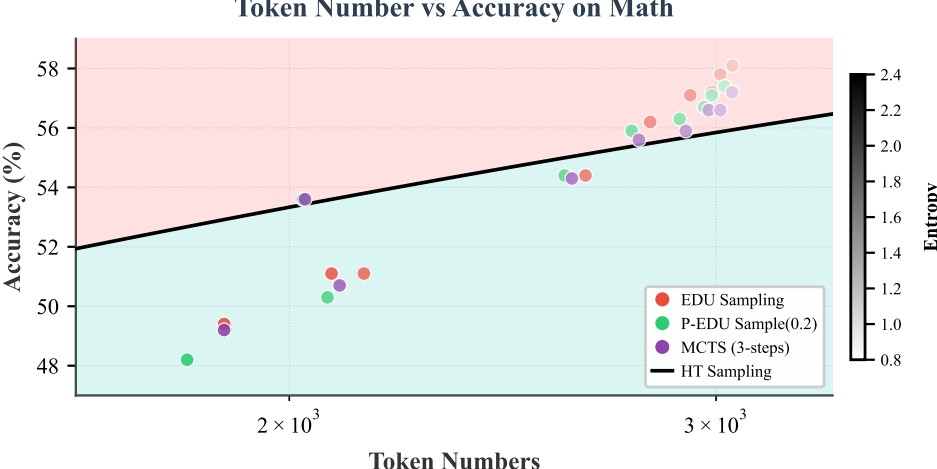

Figure 6: This figure illustrates the relationship between token count and accuracy on the MATH test set under a Max Branch Number of 8, with the performance of (High-Temperature) HT Sampling across varying token counts fitted as the baseline. On the MATH test set, most data points for both EDU Sampling and P-EDU(0.2) Sampling lie above this baseline. Notably, as the entropy threshold increases, token counts decrease alongside a corresponding drop in accuracy. Additionally, MCTS also surpasses the HT Sampling baseline when the entropy threshold is reduced.

strained by its rollout depth. When the number of rollout steps is limited, further increasing the token budget does not yield additional accuracy gains. In practice, the optimal strategy should be chosen according to the computational budget and the PRM's ability to reliably score candidate paths, with pruning used to focus resources on the most promising solution trajectories, and MCTS providing additional foresight through simulated future exploration.

### 4.7 Ablation

To further investigate the impact of decoding strategies, we introduce a variant called Sample-EDU PRM. Different from the Greedy-EDU PRM, which utilizes a deterministic greedy decoding approach, Sample-EDU PRM employs stochastic sampling (with temperature $t = 0.7$) during the decoding phase whenever no anchor is detected, while keeping all other parameters unchanged, including training methods and the base model.

Our experimental results indicate that Greedy-EDU PRM consistently achieves higher accuracy as the sample size increases (Figure 1). This improvement can be largely attributed to the deterministic nature of greedy decoding, which helps maintain reasoning consistency throughout the EDU segmentation process. When combined with entropy-thresholded branching, this method strikes a balance between solution diversity and stability, effectively avoiding the additional noise often associated with stochastic sampling.

In contrast, Sample-EDU leverages stochastic decoding to enhance diversity among candidate solutions. However, this increased diversity comes at the cost of greater variability and noise, which tends to weaken the model's inductive bias and makes performance evaluation less reliable. Overall, these findings highlight the trade-offs between diversity and consistency in reasoning, suggesting that a deterministic approach may be better suited for maintaining robust performance in EDU-PRM.

## 5 Analysis: Entropy Threshold, Accuracy, and Token Count

### 5.1 Definition and Relative Branch Depth

For a solution trace with $L$ tokens, let a branch occur at token index $d$ ($1 \leq d \leq L$). We define the relative depth as $r = \frac{d}{L}$. Aggregating $r$ across traces into a heat map (Figure 11) provides a

normalized view of where branching tends to concentrate along the trajectory. This metric serves as the foundation for our subsequent analyses on branch timing and behavior.

## 5.2 EFFECT OF ENTROPY THRESHOLD ON BRANCH TIMING

With the relative branch depth metric established, we next examine how the entropy threshold influences the timing of branch points. Figure 12 and Table 4 and Table 5 show that lowering the entropy threshold shifts branch points earlier in the sequence. A stricter threshold induces earlier branching by pruning diffuse exploratory branches, focusing the search on high-probability paths. Figure 11 further demonstrates that, under selected thresholds, EDU sampling often branches near the very start, resulting in a sharply peaked distribution of relative depths. These results indicate that entropy-based control can effectively modulate when and where branching occurs.

## 5.3 LEXICAL CHARACTERISTICS OF BRANCH NODES

Having identified where branching tends to occur, we now investigate the lexical nature of branch-point tokens. We examine the full-word forms of branch-point tokens and rank words by their branch-point frequency (Figures 8–9, MATH and OLY test sets). High-frequency items are predominantly function words (e.g., "then", "if") or light discourse operators (e.g., "thus", "so"). This observation supports our hypothesis that high-entropy tokens act as structural pivots, forming natural boundaries for controlled branching in EDU PRM. The prevalence of such words at branch points suggests that semantic structure guides the branching process.

## 5.4 ACCURACY–TOKEN TRADE-OFF

These insights into branch timing and lexical characteristics inform our understanding of the trade-offs involved in branching strategies. Figure 6 reports accuracy versus total generated tokens under varying entropy thresholds on MATH (OLY shown in Figure 13). As shown in Figure 6, lowering the entropy threshold from 2.4 to 0.8 increases accuracy from 49.4% to 58.1%, but also raises the average token count from 1,880 to 3,047 per sample. This suggests that practitioners must balance accuracy gains against computational overhead when selecting entropy thresholds. Notably, the EDU sampling begins to outperform the High-Temperature (HT) sampling only when the threshold is sufficiently low to curtail diffuse early exploration. This trade-off highlights the practical importance of threshold selection in balancing computational cost and solution quality.

Furthermore, lowering the entropy threshold tends to produce longer and more detailed reasoning paths, which may improve solution robustness but also increase resource consumption and potentially affect interpretability. Therefore, the optimal threshold may vary depending on the specific application scenario and resource constraints. Future work could explore adaptive or dynamic thresholding strategies to further enhance the efficiency and flexibility of branching methods.

## 6 CONCLUSION

We propose EDU-PRM, an entropy-guided sampling method for training process reward models that significantly advances mathematical reasoning. Our approach consistently outperforms existing baselines and, on some test sets, even matches the performance of the state-of-the-art Qwen2.5-Math-PRM. Moreover, EDU sampling improves token efficiency in solution generation. EDU-PRM demonstrates exceptional data efficiency, attaining new state-of-the-art results with minimal training data. By integrating pruning strategies like P-EDU sampling for rapid, cost-effective exploration, our framework provides complementary tools tailored to diverse task demands. Overall, EDU-PRM establishes a principled methodology for balancing accuracy, efficiency, and search depth in complex reasoning tasks, with promising avenues for future research in scaling to larger datasets, refining intermediate scoring, and developing adaptive generation strategies to extend its applicability across broader domains.

ETHICS STATEMENT

We present a technical framework that enhances model accuracy and efficiency while preserving performance integrity on publicly available models, datasets and benchmarks. No ethical or negative impacts are specifically designed in our approach, as we optimize existing models without altering their core capabilities or introducing harmful content. Our method may democratize access to advanced reasoning models by reducing computational requirements and improving data efficiency, potentially benefiting resource-constrained environments and mitigating environmental impact through more sustainable deployments.

REPRODUCIBILITY STATEMENT

We follow the standard experimental setup and details established in baselines such as Math-Shepherd and Omega PRM. For all reported results, we conduct eight experimental runs with the same random seeds and report the average performance. We use a fixed seed (1234) for the main experiments presented in the paper. Detailed experimental configurations are provided in Section 4.1. Our implementation is designed with modularity in mind, facilitating adaptation to different partial reasoning model architectures beyond those tested in this work. We will open-source our complete implementation.

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

# A    APPENDIX

## A.1    THE USE OF LARGE LANGUAGE MODELS

Large Language Models (LLMs) were used in this work solely as writing assistance tools. Specifically LLMs were employed to check for spelling errors, grammatical mistakes, and to improve the fluency and precision of expression in the paper. The LLMs did not contribute to research methodology experimental design, or data analysis. All scientific content, ideas, and conclusions presented in this paper are entirely the authors' own work.

## A.2    PROCESSBENCH

Table 1 provides a comprehensive comparison of various PRM models, including Math-Shephred, Omega, EDU variants, and Qwen-series, across three ProcessBench subsets: GSM8K, MATH, and OlympiaBench. For each dataset, we report results for both 7B and 72B model scales, including accuracy, F1 score, precision, and recall. The best performance for each metric is highlighted in bold. This detailed breakdown enables a more granular understanding of each model's strengths and limitations across different reasoning benchmarks and evaluation metrics.

## A.3    EXPERIMENTAL ENVIRONMENT, TRAINING CONFIGURATION AND DATASET DETAILS

This appendix provides detailed information on the experimental platform, framework selection, model training settings, and evaluation datasets used in this study, ensuring the reproducibility of the experiments.

### A.3.1    EXPERIMENTAL PLATFORM AND FRAMEWORK

All experiments were conducted on the **Ascend 910B platform** to ensure stable computing performance. Different frameworks were adopted for specific experimental phases to optimize efficiency:

- **PRM Training Data Production**: Employed the DeepSpeed inference framework to accelerate data processing and generation.
- **Solution Generation Phase**: Utilized the VLLM inference framework, which is optimized for high-throughput and low-latency text generation tasks.
- **PRM Training**: Adopted the Mindspeed framework, selected for its efficiency in training large-scale models for preference learning.

### A.3.2    MODEL TRAINING CONFIGURATION

Comparative experiments were conducted on two base models with different parameter scales (7B and 72B), using identical training configurations to ensure result consistency and comparability:

1. Initial learning rate: $10^{-6}$
2. Minimum learning rate (lower bound): $10^{-7}$
3. Warmup mechanism: Applied with a warmup ratio of 0.01 to stabilize parameter updates in the early training stage.
4. Cosine Annealing: Adopted a cosine strategy for subsequent learning rate adjustment, balancing late-stage convergence and overfitting prevention.
5. **Training Cycle and Checkpoint Management**:
   - Total training epochs: 5 (uniformly set for both models).
   - Checkpoint (ckpt) saving: Automatically saved at the end of each epoch to facilitate subsequent result screening and experiment reproducibility.
   - Optimal Checkpoint Selection: Compared the core metrics (e.g., accuracy, perplexity) of checkpoints from 5 epochs on the validation set; the checkpoint with the best performance was selected as the basis for final result reporting, ensuring objectivity and representativeness.

| Task | | Accuracy | F1 | Precision | Recall |
|---|---|---|---|---|---|
| **GSM8K** | | | | | |
| 7B | Math-Shephred PRM | 57.2 | 0.682 | 0.545 | 0.91 |
| | Omega PRM | 57.5 | 0.31 | 0.844 | 0.19 |
| | Sample EDU PRM | 52.5 | 0.677 | 0.513 | **0.995** |
| | Greedy EDU PRM | 55.2 | 0.218 | **0.862** | 0.125 |
| | Qwen2.5-Math-PRM-7B | **88.8** | **0.895** | 0.838 | 0.96 |
| 72B | Math-Shephred PRM | 74.5 | 0.803 | 0.671 | **1** |
| | Omega PRM | 90.5 | 0.908 | 0.882 | 0.935 |
| | Sample EDU PRM | 71 | 0.778 | 0.637 | **1** |
| | Greedy EDU PRM | 94.2 | 0.95 | 0.909 | 0.995 |
| | Qwen2.5-Math-PRM-72B | **96** | **0.961** | **0.938** | 0.985 |
| **MATH** | | | | | |
| 7B | Math-Shephred PRM | 62.9 | 0.659 | 0.615 | 0.71 |
| | Omega PRM | 58 | 0.295 | 0.917 | 0.176 |
| | Sample EDU PRM | 59.2 | 0.689 | 0.559 | **0.898** |
| | Greedy EDU PRM | 56.2 | 0.229 | **0.956** | 0.13 |
| | Qwen2.5-Math-PRM-7B | **82.4** | **0.82** | 0.839 | 0.802 |
| 72B | Math-Shephred PRM | 77.8 | 0.805 | 0.727 | 0.902 |
| | Omega PRM | 79.8 | 0.763 | **0.923** | 0.65 |
| | Sample EDU PRM | 76.4 | 0.795 | 0.709 | **0.906** |
| | Greedy EDU PRM | **88.4** | **0.882** | 0.904 | 0.862 |
| | Qwen2.5-Math-PRM-72B | 87.8 | 0.872 | 0.918 | 0.83 |
| **OlympiaBench** | | | | | |
| 7B | Math-Shephred PRM | 53.6 | 0.539 | 0.541 | 0.536 |
| | Omega PRM | 51.3 | 0.079 | 0.724 | 0.042 |
| | Sample EDU PRM | 53.8 | 0.636 | 0.528 | **0.798** |
| | Greedy EDU PRM | 51.7 | 0.083 | **0.815** | 0.004 |
| | Qwen2.5-Math-PRM-7B | **74.1** | **0.721** | 0.785 | 0.666 |
| 72B | Math-Shephred PRM | 71 | 0.74 | 0.691 | **0.796** |
| | Omega PRM | 66.1 | 0.553 | 0.816 | 0.418 |
| | Sample EDU PRM | 69.7 | 0.723 | 0.67 | 0.786 |
| | Greedy EDU PRM | 77.2 | 0.762 | 0.801 | 0.726 |
| | Qwen2.5-Math-PRM-72B | **79.8** | **0.779** | **0.86** | 0.712 |

Table 1: Performance comparison of different PRM models (Math-Shephred, Omega, EDU, Qwen-series) on three ProcessBench subsets: GSM8K, MATH, and OLY. For each dataset, results are reported for both 7B and 72B model sizes, including metrics for accuracy, F1 score, precision, and recall. The best results for each metric are highlighted in bold.

### A.3.3 DETAILS OF EVALUATION DATASETS

Five datasets covering different difficulty levels (from elementary to university-level) and task types (math reasoning, multi-step problem-solving) were used to comprehensively evaluate the model's generalization and reasoning abilities. The key details of each dataset are presented in Table 2.

### A.4 EDU SAMPLING WHITELIST

\, $, \n, \r, ␣, _, ␣␣, :, \(, \[, \{, ␣, \], \), \}, \[, \(, (, \[, [, \], \{, {, \}, }

| Dataset | Description | Usage in Evaluation |
|---|---|---|
| OlympiadBench | Bilingual, multimodal dataset with 8,952 math/physics questions (from Olympiads, college entrance exams); subset "OE_TO_maths_en_COMP" contains 675 problems. | Used the "OE_TO_maths_en_COMP" subset (675 problems) to evaluate the model's performance on competitive/advanced math tasks. |
| GSM8K | 8,500+ grade school math word problems (linguistically diverse, requiring 2–8 steps of basic arithmetic reasoning); solutions in natural language; 1,319 test data points. | Used 1,319 test data points to evaluate the model's elementary mathematical reasoning and multi-step natural language-based problem-solving skills. |
| MATH | Consists of 12,500 challenging competition-level mathematics problems, each with detailed step-by-step solutions. We selected 5,000 problems as our test set to evaluate the model's abilities in complex mathematical reasoning, solution derivation , and answer generation. The MATH dataset serves as a rigorous benchmark for assessing advanced mathematical problem-solving skills. | Used the selected 5,000-test-sample subset to systematically evaluate the model's reasoning process, step-by-step solution generation, and overall accuracy on advanced math problems. |
| CollegeMath | ~1100 university-level math problems (covering 6 college math areas; 20% with images). | Used all test data to assess the model's proficiency in complex, advanced mathematical concepts (relevant to industry and higher education scenarios). |
| ProcessBench | Three selected subsets: MATH (1,000 samples), OlympiaBench (1,000 samples), GSM8K (400 samples); each sample includes step-by-step error position annotations and final solution correctness labels; balanced positive/negative samples in each subset. | Used to evaluate the model's overall solution correctness. |

Table 2: Key details of evaluation datasets used in our experiments.

## A.5 EVALUATION PROMPT

We use the following prompt to evaluate the solution, with Qwen3-32B-instruct (Yang et al., 2025) as the underlying model. For each test instance, the model is provided with the problem statement and instructed to generate a step-by-step solution. The prompt is designed to encourage detailed reasoning and explicit justification at each step, ensuring the model's output is both accurate and interpretable.

---

You are a high school math teacher who is correcting homework. You need to check whether the result from Response is consistent with the Answer. If the result is consistent, you should reply Yes, otherwise you should reply No. You only need to compare the result from Response to the Answer. If the content in the Response does not give the final result, answer No.

Here are some examples

**[Question]**

If $x = 3$ and $y = 2$, then what is the value of $\dfrac{2x^3 - 3y^2}{6}$ ?

**[Response]**

To find the value of the expression $\dfrac{2x^3 - 3y^2}{6}$ given $x = 3$ and $y = 2$, we substitute the values of $x$ and $y$ into the expression and simplify step by step.

---

1. **Substitute $x = 3$ and $y = 2$:**

$$\frac{2(3)^3 - 3(2)^2}{6}$$

2. **Simplify the expression inside the numerator:**

$$3^3 = 27, \qquad 2^2 = 4$$

So,

$$\frac{2(27) - 3(4)}{6}$$

3. **Further simplify:**

$$2 \times 27 = 54, \qquad 3 \times 4 = 12$$

Hence

$$\frac{54 - 12}{6}$$

4. **Subtract in the numerator:**

$$54 - 12 = 42 \Rightarrow \frac{42}{6}$$

5. **Simplify the fraction:**

$$42 \div 6 = 7$$

Therefore the value is $\boxed{7}$.

**[Answer]**
7
**[Evaluation]**
Yes
**[Question]**
Find $\cos^{-1}\left(\frac{1}{2}\right)$. Express your answer in radians.
**[Response]**
To find $\cos^{-1}\left(\frac{1}{2}\right)$, we need the angle whose cosine is $\frac{1}{2}$. From the 30-60-90 triangle, $\cos\left(\frac{\pi}{6}\right) = \frac{1}{2}$. Therefore $\cos^{-1}\left(\frac{1}{2}\right) = \frac{\pi}{6}$. So the answer is $\boxed{\frac{\pi}{6}}$.
**[Answer]**
$\frac{\pi}{3}$
**[Evaluation]**
No
**[Question]**
Consider two lines: line $l$ parameterized as

$$x = 1 + 4t, \qquad y = 4 + 3t$$

and the line $m$ parameterized as

$$x = -5 + 4s, \qquad y = 6 + 3s.$$

Let $A$ be a point on line $l$, $B$ be a point on line $m$, and let $P$ be the foot of the perpendicular from $A$ to line $m$.

Then $\overrightarrow{BP}$ is the projection of $\overrightarrow{BA}$ onto some vector $\begin{pmatrix} v_1 \\ v_2 \end{pmatrix}$ such that $v_1 + v_2 = -7$. Find $\begin{pmatrix} v_1 \\ v_2 \end{pmatrix}$.
**[Response]**
(Working leading to)

$$\boxed{\begin{pmatrix} -4 \\ -3 \end{pmatrix}}$$

**[Answer]**
$$\begin{pmatrix} -4 \\ -3 \end{pmatrix}$$
**[Evaluation]**
Yes
**[Question]**
Consider two lines: line $l$ parameterized as

$$x = 1 + 4t, \qquad y = 4 + 3t$$

and the line $m$ parameterized as

$$x = -5 + 4s, \qquad y = 6 + 3s.$$

Let $A$ be a point on line $l$, $B$ be a point on line $m$, and let $P$ be the foot of the perpendicular from $A$ to line $m$.
Then $\overrightarrow{BP}$ is the projection of $\overrightarrow{BA}$ onto some vector $\begin{pmatrix} v_1 \\ v_2 \end{pmatrix}$ such that $v_1 + v_2 = -7$. Find $\begin{pmatrix} v_1 \\ v_2 \end{pmatrix}$.
**[Response]**
(An unrelated distance-to-plane calculation producing 4.)
**[Answer]**
$\dfrac{10}{3}$
**[Evaluation]**
No
Note: You only need to compare the result from Response to the Answer.
**[Question]**
⟪ question ⟫
**[Response]**
⟪ Response ⟫
**[Answer]**
⟪*correctanswer*⟫
**[Evaluation]**

### A.6 COMPARISON OF PRMS

Table 3 presents a comprehensive comparison of various PRMs across four benchmark datasets: OLY, MATH, GSM8K, and Collegemath. The models evaluated include Qwen2.5-Math-PRM, Math-Shepherd (ours), Omega, Sample-EDU, and EDU, with parameter sizes ranging from 7B to 72B. For each dataset, models are grouped according to their parameter sizes to facilitate a fair comparison. The evaluation is conducted under different sample sizes (2, 4, 8, 16, 32, 64, and 128), allowing for an analysis of performance scaling as the sample size increases. Bolded values in the table highlight the best-performing model for each sample size within the respective dataset. This table serves as a supplementary resource for section 4.4.

### A.7 PERFORMANCE COMPARISON OF EDU-BASED SAMPLE METHODS

Table 4 and Table 5 summarize the performance of EDU sampling, P-EDU, and MCTS-EDU methods on the MATH and OLY datasets, respectively, under varying entropy thresholds with a fixed maximum branch number of 8. Each table reports both the accuracy (%) and the average number of tokens consumed for each method and entropy setting.

The results illustrate several key trends:

- For both datasets, increasing the entropy threshold generally leads to a reduction in average token usage, but this is often accompanied by a decrease in accuracy.

- The P-EDU Sampling, which incorporates entropy-based pruning, can sometimes outperform the standard EDU Sampling depending on the underlying PRM's ability to identify and prune low-confidence branches.

- The accuracy improvement of MCTS-EDU is constrained by the rollout depth; with limited rollout steps, its accuracy does not continue to increase with higher token counts.

These tables provide a comprehensive overview of how entropy-based branching and pruning strategies affect the balance between accuracy and token efficiency across different reasoning methods.

### A.8 COMPREHENSIVE COMPARISON OF EDU SAMPLING ON MATH AND OLY DATASETS BY DIFFERENT MAXIMUM BRANCH

Table 6 presents a detailed comparison of several branching strategies—HT Sampling, EDU Sampling, P-EDU Sampling, and MCTS Sampling—on both the MATH and OLY datasets as the maximum allowed number of branches varies from 1 to 64. The table includes three main metrics: accuracy (%) using the 72B model, total tokens consumed (in millions), and average tokens per problem for each method and branch setting.

Key observations include:

- Increasing the maximum branch number generally leads to higher accuracy for most methods, but also significantly increases token usage.

- EDU Sampling and P-EDU Sampling demonstrate better token efficiency compared to HT Sampling, especially at higher branch limits.

- MCTS Sampling's accuracy plateaus or even drops at higher branch numbers, but its token usage remains relatively low due to its targeted search mechanism.

- OLY dataset results show lower overall accuracy compared to MATH, but similar scaling trends in token usage and efficiency.

This table provides a comprehensive overview of how different branching and sampling strategies scale with computational resources, highlighting the trade-offs between accuracy gains and token consumption.

### A.9 MULTI-LEVEL PRUNING IMPACT ON PRM SCORE DISTRIBUTION

This figure 7 illustrates the effects of multi-level threshold-based pruning on PRM scores for a large model. The visualization covers six pruning levels (from 1 to 6), showing how the distribution of PRM scores changes as nodes are either retained or deleted. For each level, the panels display the cumulative distribution functions (CDFs) comparing retained and deleted nodes, as well as frequency histograms indicating their counts. Additionally, the mean PRM scores for both groups are presented, providing insight into the impact of pruning on model performance and node characteristics.

### A.10 WORD FREQUENCY ANALYSIS ACROSS DATASETS AND BRANCH CONFIGURATIONS

Figure 8 presents word cloud visualizations for the MATH and OLY datasets under different entropy conditions, with the maximum branch number set to 8. In these visualizations, the size of each word corresponds to its frequency within the dataset, allowing for an intuitive comparison of commonly used terms across different entropy settings.

Figure 9 shows word cloud visualizations for OLY and MATH samples under varying maximum branch numbers. The font size of each word indicates its frequency, with larger fonts representing words that appear more frequently in the samples. These figures provide insights into the distribution of key terms in educational samples, highlighting differences in word usage patterns across datasets and branching configurations.

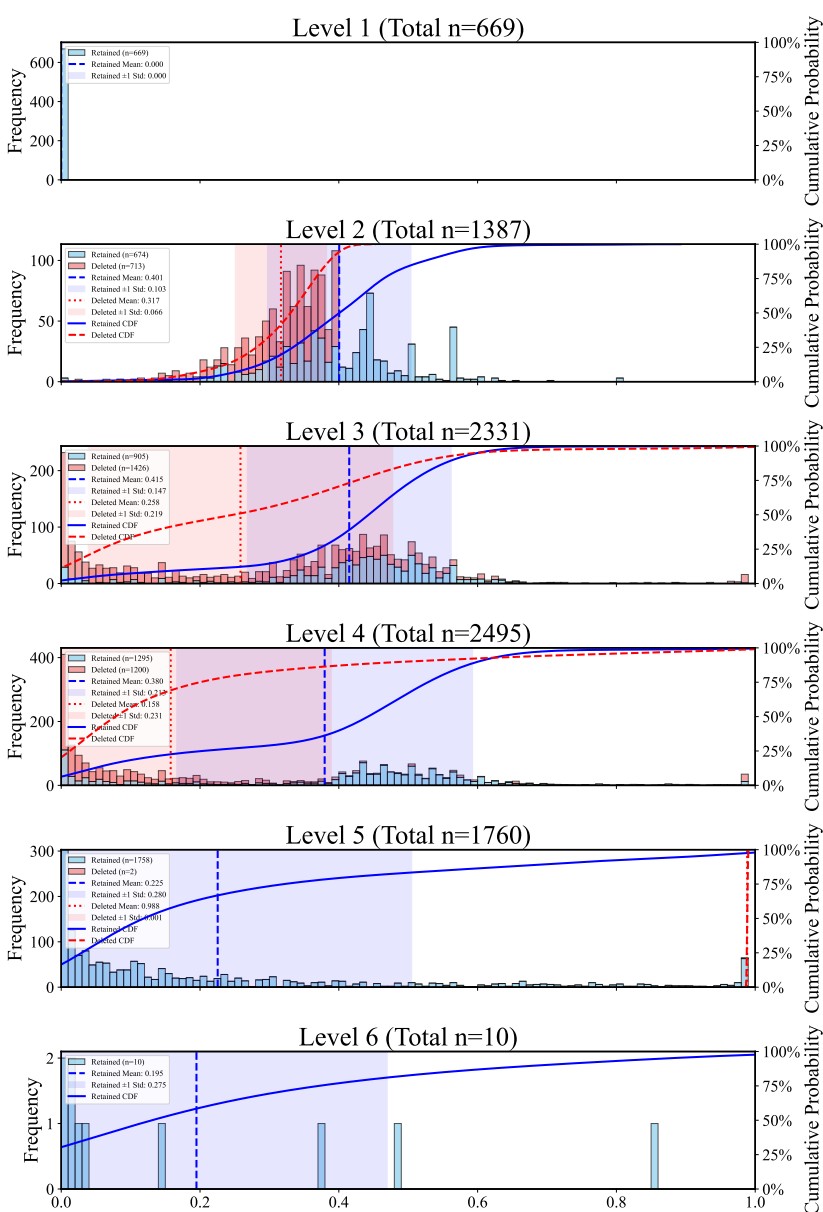

Figure 7: Multi-level Pruning Effects on PRM. This visualization presents the distribution of PRM scores across six levels (1 to 6) for a large model, illustrating the effect of threshold-based pruning on node retention and deletion. Each panel includes a cumulative distribution function (CDF) comparing retained and deleted nodes, along with frequency histograms showing their counts, and displays the mean PRM scores for both groups.

## A.11 ILLUSTRATIVE EXAMPLE OF AN EDU SAMPLING

Figure 10 presents a real example of an EDU Sampling, illustrating the process of branch selection and token evaluation. In this example, a specific branch is highlighted for clarity. The segments shown in red represent tokens whose entropy values fall below the predefined threshold, indicating

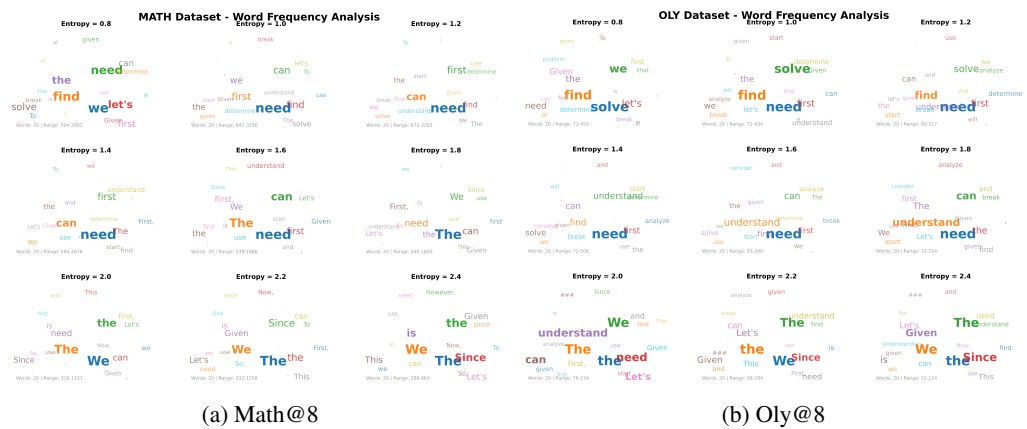

(a) Math@8                                    (b) Oly@8

Figure 8: Word cloud visualizations for the MATH and OLY datasets under different entropy conditions by EDU Sampling, where the maximum branch number is set to 8. The size of each word reflects its frequency in the dataset, with more frequent words shown in larger font.

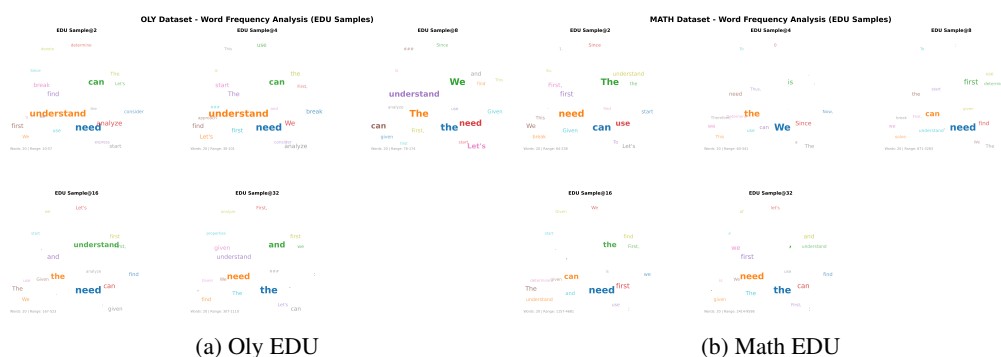

(a) Oly EDU                                    (b) Math EDU

Figure 9: Word cloud visualizations for Oly and MATH samples under different maximum branch numbers by EDU Sampling. The font size of each word indicates its frequency, with more frequently occurring words shown in larger font.

points of higher confidence during the reasoning process. At each step, the Label is determined through backpropagation from the final solution outcome, providing insight into the contribution of each token to the overall result. This visualization demonstrates how entropy-based selection and backpropagation labeling work together to guide the reasoning trajectory in the EDU Sampling framework.

## A.12  HEATMAP ANALYSIS OF NODE BRANCH POINT DISTRIBUTIONS

Figure 11 and Figure 12 provide heatmap visualizations of node and branch point distributions under different experimental conditions on the OLY and MATH test sets.

Figure 11 shows the concentration of nodes within the initial 0–20% interval of solution steps for varying Maximum Branch Number settings. Red regions indicate a higher concentration of nodes, while blue regions represent lower concentrations. Compared to MATH, the OLY test set displays a more front-loaded distribution, with nodes concentrated earlier in the solution process.

Figure 12 illustrates branch point distributions at a fixed Maximum Branch Number of 8 under different entropy thresholds, focusing on the 1–20% segment. Lower entropy thresholds result in earlier branching, and for any given threshold, OLY consistently shows branch points occurring earlier than MATH. These observations highlight structural differences in reasoning trajectories and branching dynamics between the two datasets.

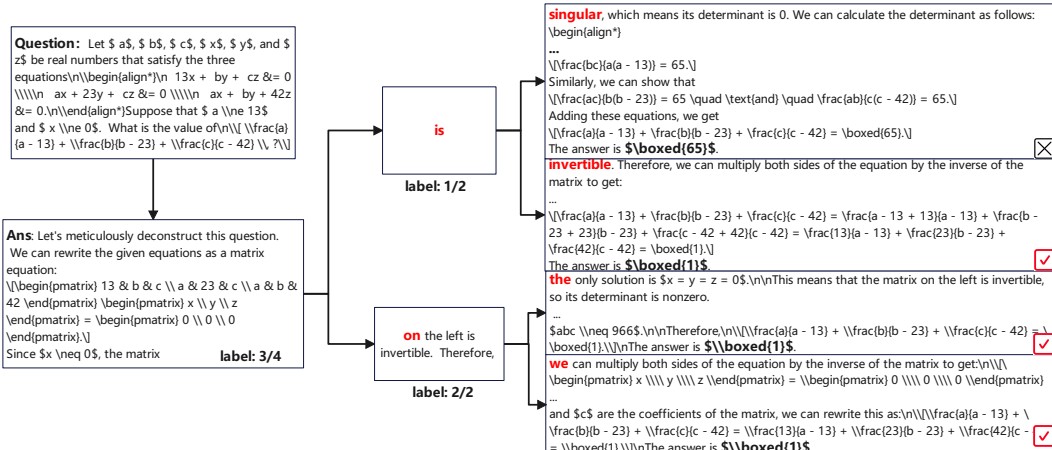

Figure 10: This is a real example of an EDU Sampling, where a selected branch is presented for illustration. The red-colored segments correspond to tokens with entropy values below the predefined threshold. For each step, the Label is derived from the results obtained through backpropagation based on the final outcome.

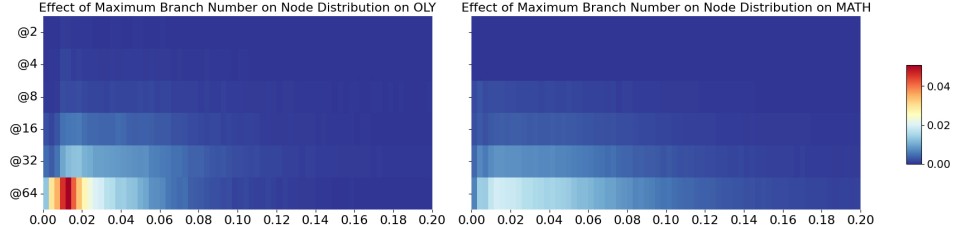

Figure 11: Heatmaps of node distribution under different Maximum Branch Number settings on the OLY and MATH test sets, restricted to the 0–20% interval of solutions. Red denotes a higher concentration of nodes in that percentile range, whereas blue denotes a lower concentration. Relative to MATH, OLY exhibits a more front-loaded (early-range) concentration.

## A.13 TOKEN COUNT VS. ACCURACY ANALYSIS ACROSS SAMPLING METHODS WITH DIFFERENT ENTROPY

Figure 13 illustrates the relationship between token count and accuracy on the OlympiaBench and MATH test sets under a Max Branch Number of 8. The performance of HT Sampling across different token counts is fitted as the baseline for comparison. On the MATH test set, most data points for both

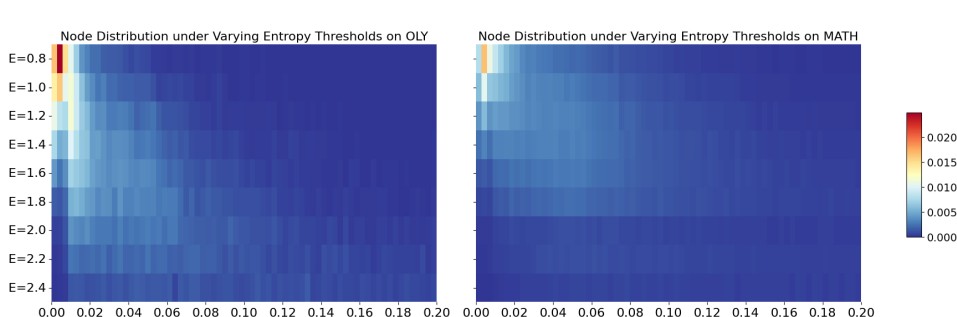

Figure 12: Heatmaps of branch point distribution on the OLY and MATH test sets at a Max Branch Number of 8 under varying entropy thresholds (only the 1–20% segment shown). Lower entropy thresholds trigger earlier branching, and for any fixed threshold, OLY exhibits earlier branch points than MATH.

EDU Sampling and P-EDU(0.2) Sampling are positioned above this baseline, indicating superior performance in terms of accuracy relative to token count. As the entropy threshold increases, the number of tokens required decreases, but this reduction is accompanied by a corresponding drop in accuracy. Additionally, the MCTS method also exceeds the HT Sampling baseline when the entropy threshold is set lower, further highlighting the impact of entropy-based branching strategies on solution efficiency and accuracy.

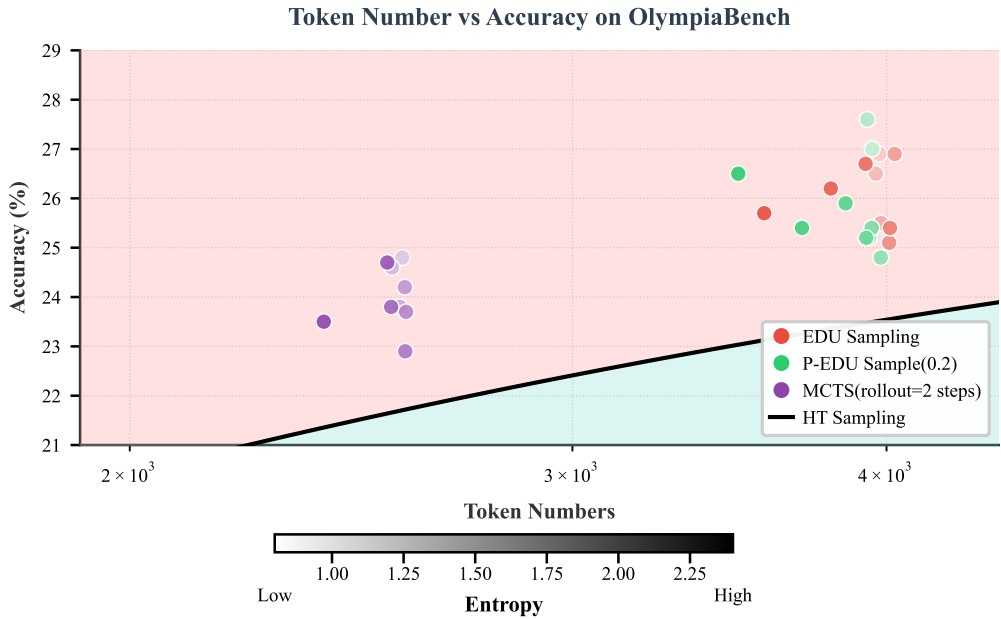

Figure 13: This figure illustrates the relationship between token count and accuracy on the OlympiaBench test set under a Max Branch Number of 8, with the performance of HT Sampling across varying token counts fitted as the baseline. On the MATH test set, most data points for both EDU Sampling and P-EDU(0.2) Sampling lie above this baseline. Notably, as the entropy threshold increases, token counts decrease alongside a corresponding drop in accuracy.

| Datasets | Models | Samples | | | | | | |
|---|---|---|---|---|---|---|---|---|
| | | 2 | 4 | 8 | 16 | 32 | 64 | 128 |
| **OLY** | Math-Shepherd-Mistral-7B-PRM | 15.9 | 16.3 | 17.5 | 17.6 | 18.2 | 18.8 | 17.9 |
| | Qwen2.5-Math-7B-PRM800K | 16 | 18.2 | 19.3 | 19.9 | 20.3 | 21.3 | 22.7 |
| | Qwen2.5-Math-PRM-7B | **17.9** | **20.7** | **23** | **23.6** | **24.6** | **25.8** | **28.9** |
| | Math-Shephred-7B | 16.9 | 16.4 | 15.1 | 15.1 | 15.4 | 13.9 | 13.8 |
| | Omega-7B | 14.5 | 15.3 | 16 | 17.5 | 17.5 | 16.9 | 17.9 |
| | Sample-EDU-7B | 17.5 | 18.1 | 18.7 | 18.2 | 19.1 | 19.1 | 20.1 |
| | EDU-7B | 16 | 19.4 | 18.4 | 18.2 | 19.7 | 19.4 | 20 |
| | Qwen2.5-Math-RM-72B | **19.4** | 21.8 | 24.4 | 25.5 | 27.4 | 29.2 | 30.4 |
| | Qwen2.5-Math-PRM-72B | 18.8 | 21.9 | 24.7 | 25.8 | 27 | 28.6 | 29.3 |
| | Math-Shephred-72B | 18.8 | 20.4 | 21.9 | 22.4 | 23.6 | 24.7 | 26.7 |
| | Omega-72B | 18.7 | 20.7 | 21.1 | 22.5 | 24.6 | 24.4 | 25.5 |
| | Sample-EDU-72B | 18.8 | 21 | 22.2 | 22.4 | 23.6 | 24.1 | 27 |
| | EDU-72B | **19.4** | **22.4** | **25.5** | **26.7** | **27.6** | **30.2** | **32.7** |
| **MATH** | Math-Shepherd-Mistral-7B-PRM | 43.7 | 45.0 | 45.6 | 46.3 | 46.5 | 46.2 | 46.5 |
| | Qwen2.5-Math-7B-PRM800K | 45.8 | 48.2 | 50.1 | 50.7 | 51 | 51.2 | 51 |
| | Qwen2.5-Math-PRM-7B | **47.4** | **51.3** | **54.8** | **58.2** | **60.9** | **62.5** | **64.6** |
| | Math-Shephred-7B | 43.8 | 44.8 | 45.2 | 45.5 | 46.2 | 46.2 | 46.1 |
| | Omega-7B | 43.4 | 43.7 | 44.5 | 45.6 | 46.8 | 47.6 | 48.5 |
| | Sample-EDU-7B | 44 | 46.5 | 47.6 | 48.4 | 49.7 | 50.1 | 50.4 |
| | EDU-7B | 44 | 46.3 | 47.7 | 48.9 | 49.6 | 50.6 | 51.3 |
| | Qwen2.5-Math-RM-72B | 48.6 | **54** | **57.8** | **62.0** | **65.4** | **67.9** | **70.0** |
| | Qwen2.5-Math-PRM-72B | 47.2 | 51.5 | 54.8 | 57.9 | 60.5 | 61.7 | 63.6 |
| | Math-Shephred-72B | 47 | 50.9 | 54.4 | 57.1 | 59 | 60.4 | 61.7 |
| | Omega-72B | 48 | 52.1 | 54.7 | 57.4 | 59.7 | 61.4 | 62.4 |
| | Sample-EDU-72B | 46.9 | 50.4 | 53.8 | 56.5 | 58.8 | 60.3 | 61.8 |
| | EDU-72B | **48.9** | 53.9 | 57.2 | 61.3 | 62.9 | 64.7 | 65.5 |
| **GSM8K** | Math-Shepherd-Mistral-7B-PRM | 84.7 | 85.2 | 85.4 | 86 | 84.7 | 84.8 | 84.8 |
| | Qwen2.5-Math-7B-PRM800K | 84.3 | 86.1 | 87 | 87.2 | 87.6 | 88.1 | 87.8 |
| | Qwen2.5-Math-PRM-7B | **85.6** | **87** | **88.6** | **88.6** | **88.9** | **89.3** | **89.3** |
| | Math-Shephred-7B | 83.3 | 83 | 83.2 | 83.4 | 83 | 83.1 | 82.6 |
| | Omega-7B | 82.9 | 83.2 | 83.4 | 83.7 | 85 | 85 | 85.7 |
| | Sample-EDU-7B | 82.6 | 82.5 | 82.3 | 82.6 | 83 | 83.4 | 83.5 |
| | EDU-7B | 83.9 | 84 | 83.7 | 84.8 | 85.4 | 86.5 | 86.7 |
| | Qwen2.5-Math-RM-72B | **87.3** | 89.7 | **91.1** | **91.9** | **92.3** | **92.6** | **92.7** |
| | Qwen2.5-Math-PRM-72B | 86.4 | 87.7 | 88.7 | 88.9 | 89.3 | 89.9 | 90.3 |
| | Math-Shephred-72B | 86.1 | 87.6 | 88.3 | 88.1 | 88 | 88.6 | 89.5 |
| | Omega-72B | 85.4 | 86.3 | 87.6 | 88.6 | 89.2 | 90 | 90.1 |
| | Sample-EDU-72B | 85.5 | 87.1 | 87.6 | 87.6 | 87.9 | 88.2 | 88.1 |
| | EDU-72B | 87 | **89.8** | 90.6 | 91.8 | 92.1 | 92 | 91.5 |
| **Collegemath** | Math-Shepherd-Mistral-7B-PRM | 11.8 | 11.8 | 11.8 | 11.6 | 11.7 | 11.8 | 11.8 |
| | Qwen2.5-Math-7B-PRM800K | 11.7 | 11.9 | 11.8 | 11.6 | 11.6 | 11.5 | 11.6 |
| | Qwen2.5-Math-PRM-7B | **11.9** | **12.3** | **12.7** | **13.0** | **13.2** | **13.6** | **14.1** |
| | Math-Shephred-7B | 11.5 | 11.8 | 11.9 | 11.9 | 11.8 | 11.9 | 11.9 |
| | Omega-7B | 11.7 | 11.6 | 11.7 | 11.8 | 12 | 11.9 | 12.1 |
| | Sample-EDU-7B | 11.6 | 12 | 12 | 12.3 | 12.3 | 12.5 | 12.6 |
| | EDU-7B | 11.6 | 11.7 | 11.6 | 11.6 | 12.1 | 12 | 12.2 |
| | Qwen2.5-Math-RM-72B | 12.1 | 12.6 | 13.3 | 13.9 | **14.5** | **15.1** | **15.7** |
| | Qwen2.5-Math-PRM-72B | 12 | 12.3 | 12.6 | 12.9 | 13.1 | 13 | 13.2 |
| | Math-Shephred-72B | 12 | 12.5 | 13.2 | 13.8 | 13.8 | 14.3 | 14.8 |
| | Omega-72B | 12 | 12.4 | 13.2 | 13.5 | 13.9 | 14.3 | 14.8 |
| | Sample-EDU-72B | 11.8 | 12.5 | 12.9 | 13.4 | 13.7 | 14.1 | 14.5 |
| | EDU-72B | **12.3** | **12.9** | **13.4** | **14.1** | 14.4 | 14.9 | 15.5 |

Table 3: Comparison of performance across different datasets (OLY, MATH, GSM8K, and College-math) and various PRMs (including Qwen2.5-Math-PRM, Math-Shephred (ours), Omega, Sample-EDU, and EDU with 7B and 72B parameters, Qwen2.5-Math-7B-PRM800K, Qwen2.5-Math-72B-PRM, Math-Shepherd-Mistral-7B-PRM) under different sample sizes (2, 4, 8, 16, 32, 64, and 128). Models are grouped by parameter size within each dataset. The **bold** values indicate the highest performance score in each column for the corresponding dataset, and the underlined values denote the second highest score.

| Method | Entropy | | | | | | | | |
|---|---|---|---|---|---|---|---|---|---|
| | 0.8 | 1.0 | 1.2 | 1.4 | 1.6 | 1.8 | 2.0 | 2.2 | 2.4 |
| EDU-7B | 47.7 | 47.8 | 47.5 | 47.2 | 46.1 | 46.0 | 45.7 | 42.8 | 42.0 |
| EDU-72B | 58.1 | 57.8 | 57.2 | 57.1 | 56.2 | 54.4 | 51.1 | 51.1 | 49.4 |
| P-EDU-0.2 | 57.4 | 57.1 | 56.7 | 56.3 | 55.9 | 54.4 | 53.6 | 50.3 | 48.2 |
| P-EDU-0.3 | 55.6 | 55.5 | 55.5 | 55.1 | 55.2 | 53.8 | 53.2 | 49.8 | 48.6 |
| P-EDU-0.4 | 52.2 | 52.7 | 53.5 | 52.4 | 53.1 | 52.0 | 52.5 | 48.9 | 48.0 |
| MCTS-EDU (1-step) | 48.7 | 48.8 | 48.3 | 48.7 | 47.9 | 46.7 | 48.7 | 45.6 | 45.5 |
| MCTS-EDU (2-step) | 53.2 | 53.2 | 53.6 | 52.9 | 52.5 | 52.2 | 51.8 | 48.7 | 47.8 |
| MCTS-EDU (3-step) | 57.2 | 56.6 | 56.6 | 55.9 | 55.6 | 54.3 | 53.6 | 50.7 | 49.2 |
| *EDU Average Token* | 3047 | 3012 | 2988 | 2927 | 2818 | 2650 | 2082 | 2147 | 1880 |
| *P-EDU-0.2 Average Token* | 3024 | 2988 | 2966 | 2898 | 2769 | 2598 | 2026 | 2074 | 1815 |
| *P-EDU-0.3 Average Token* | 2434 | 2533 | 2611 | 2610 | 2537 | 2393 | 1904 | 1935 | 1705 |
| *P-EDU-0.4 Average Token* | 1711 | 1780 | 1875 | 1888 | 1896 | 1835 | 1594 | 1577 | 1405 |
| *MCTS-EDU (1-step) Average Token* | 1026 | 1010 | 1009 | 997 | 998 | 975 | 937 | 920 | 869 |
| *MCTS-EDU (2-step) Average Token* | 1863 | 1849 | 1834 | 1818 | 1782 | 1710 | 1464 | 1482 | 1347 |
| *MCTS-EDU (3-step) Average Token* | 3046 | 3012 | 2979 | 2915 | 2788 | 2616 | 2030 | 2098 | 1880 |

Table 4: Accuracy and Average Token Usage of EDU Sampling, P-EDU, and MCTS-EDU Methods on the MATH Dataset Across Different Entropy Thresholds (Max Branches = 8). Higher entropy values correspond to later branching and fewer tokens. The table reports both accuracy (%) and average token count for each method and threshold.

| Method | Entropy | | | | | | | | |
|---|---|---|---|---|---|---|---|---|---|
| | 0.8 | 1.0 | 1.2 | 1.4 | 1.6 | 1.8 | 2.0 | 2.2 | 2.4 |
| EDU-7B | 21.5 | 20.8 | 20.0 | 18.8 | 18.3 | 20.0 | 21.3 | 20.0 | 19.4 |
| EDU-72B | 26.9 | 26.5 | 25.5 | 26.9 | 25.1 | 25.4 | 26.7 | 26.2 | 25.7 |
| P-EDU-0.2 | 27.0 | 27.6 | 25.2 | 24.8 | 25.4 | 25.2 | 25.9 | 25.4 | 26.5 |
| P-EDU-0.3 | 25.5 | 26.4 | 24.4 | 24.2 | 24.2 | 24.6 | 25.6 | 24.7 | 25.8 |
| P-EDU-0.4 | 23.3 | 24.1 | 22.5 | 22.1 | 23.1 | 22.2 | 25.1 | 24.4 | 24.4 |
| MCTS-EDU (1-step) | 21.8 | 22.8 | 20.6 | 21.6 | 21.0 | 20.2 | 21.7 | 20.2 | 21.7 |
| MCTS-EDU (2-step) | 24.8 | 24.6 | 23.8 | 24.2 | 23.7 | 22.9 | 23.8 | 24.7 | 23.5 |
| MCTS-EDU (3-step) | 26.0 | 26.1 | 24.3 | 24.5 | 24.3 | 24.6 | 25.1 | 24.9 | 25.0 |
| *EDU Average Token* | 3973 | 3961 | 3980 | 4030 | 4010 | 4013 | 3924 | 3801 | 3576 |
| *P-EDU-0.2 Average Token* | 3948 | 3930 | 3937 | 3979 | 3946 | 3926 | 3853 | 3702 | 3492 |
| *P-EDU-0.3 Average Token* | 3122 | 3227 | 3352 | 3417 | 3474 | 3488 | 3499 | 3399 | 3236 |
| *P-EDU-0.4 Average Token* | 2260 | 2721 | 2844 | 2916 | 2962 | 3016 | 3082 | 3095 | 2936 |
| *MCTS-EDU (1-step) Average Token* | 1449 | 1430 | 1437 | 1437 | 1451 | 1428 | 1432 | 1388 | 1347 |
| *MCTS-EDU (2-step) Average Token* | 2567 | 2543 | 2561 | 2573 | 2576 | 2574 | 2541 | 2532 | 2389 |
| *MCTS-EDU (3-step) Average Token* | 2972 | 3961 | 3981 | 4025 | 4014 | 4009 | 3909 | 3792 | 3547 |

Table 5: Accuracy (%) Comparison of EDU Sampling, P-EDU Sampling, and MCTS-EDU on OLY Dataset under Different Entropy Values (Max Branches = 8)

| Method | MATH Dataset | | | | | | | OLY Dataset | | | | | | |
|---|---|---|---|---|---|---|---|---|---|---|---|---|---|---|
| | **1** | **2** | **4** | **8** | **16** | **32** | **64** | **1** | **2** | **4** | **8** | **16** | **32** | **64** |
| **Performance (%) - 72B Model** | | | | | | | | | | | | | | |
| HT Sampling | 42.2 | 48.9 | 53.9 | 57.2 | 61.3 | 62.9 | 64.7 | 14.2 | 19.4 | 22.4 | 25.5 | 26.7 | 27.6 | 30.2 |
| EDU Sampling | 41.8 | 50.7 | 55.0 | 57.4 | 62.4 | 64.7 | 67.3 | 20.2 | 21.7 | 24.8 | 26.7 | 28.9 | 31.7 | **33.2** |
| P-EDU (0.2) | 41.8 | 46.3 | 51.1 | 57.1 | 60.8 | 63.2 | 65.2 | 20.2 | 21.5 | 25.1 | 25.9 | 28.8 | 32.1 | 32.2 |
| P-EDU (0.3) | 41.8 | 46.3 | 51.1 | 55.5 | 59.7 | 61.8 | 63.7 | 20.2 | 21.5 | 24.7 | 25.6 | 28.1 | 30.9 | 30.0 |
| P-EDU (0.4) | 41.8 | 46.3 | 50.8 | 52.7 | 56.0 | 57.4 | 59.2 | 20.2 | 21.5 | 23.1 | 25.1 | 24.4 | 26.2 | 27.8 |
| MCTS (1) | 41.8 | 46.3 | 50.4 | 48.8 | 48.6 | 47.6 | 47.8 | 20.2 | 21.5 | 22.7 | 21.7 | 20.5 | 21.2 | 22.1 |
| MCTS (2) | 41.8 | 46.3 | 51.1 | 53.2 | 53.7 | 54.2 | 53.4 | 20.2 | 21.5 | 25.3 | 23.8 | 23.1 | 23.0 | 25.5 |
| MCTS (3) | 41.8 | 46.3 | 51.2 | 56.6 | 57.2 | 55.9 | 56.8 | 20.2 | 21.5 | 25.3 | 25.1 | 25.0 | 24.8 | 26.4 |
| **Token Usage Statistics** | | | | | | | | | | | | | | |
| *Total Tokens (M)* | | | | | | | | | | | | | | |
| HT Sampling | 2.65 | 5.28 | 10.7 | 21.7 | 43.3 | 86.5 | 173 | 0.58 | 1.12 | 2.23 | 4.45 | 8.92 | 17.9 | 35.7 |
| EDU Sampling | 0.49 | 0.93 | 1.80 | 3.66 | 7.38 | 14.8 | 29.9 | 0.49 | 0.93 | 1.80 | 3.66 | 7.38 | 14.8 | 29.9 |
| *Average Tokens per Problem* | | | | | | | | | | | | | | |
| BON Sampling | 530 | 1,056 | 2,146 | 4,338 | 8,650 | 17,306 | 34,623 | 853 | 1,655 | 3,298 | 6,591 | 13,213 | 26,489 | 52,848 |
| EDU Sampling | 511 | 700 | 946 | 2,988 | 5,980 | 11,882 | 23,546 | 643 | 1,107 | 2,034 | 3,749 | 7,153 | 15,050 | 30,524 |
| P-EDU (0.2) | 511 | 700 | 937 | 2,031 | 3,777 | 7,753 | 22,867 | 643 | 1,107 | 2,034 | 3,930 | 7,570 | 15,050 | 30,524 |
| P-EDU (0.3) | 511 | 700 | 919 | 1,908 | 3,415 | 6,824 | 15,174 | 643 | 1,107 | 1,938 | 3,227 | 6,365 | 11,710 | 18,565 |
| P-EDU (0.4) | 511 | 700 | 874 | 1,597 | 2,569 | 4,591 | 6,896 | 643 | 1,107 | 1,660 | 2,323 | 3,804 | 5,827 | 8,540 |
| MCTS (1) | 511 | 700 | 787 | 936 | 933 | 955 | 1,053 | 643 | 1,107 | 1,339 | 1,432 | 1,475 | 1,480 | 1,489 |
| MCTS (2) | 511 | 700 | 639 | 1,465 | 1,666 | 1,681 | 2,038 | 643 | 1,107 | 2,046 | 2,541 | 2,762 | 2,825 | 2,931 |
| MCTS (3) | 511 | 700 | 946 | 2,037 | 2,633 | 2,959 | 3,963 | 643 | 1,107 | 2,048 | 3,909 | 4,932 | 5,423 | 5,683 |

Table 6: Accuracy and Token Usage Statistics for HT Sampling, EDU Sampling, P-EDU Sampling, and MCTS Sampling across Different Maximum Branch Numbers (1–64) on the MATH and OLY Datasets. The table reports accuracy (%) for the 72B model, total tokens consumed (in millions), and average tokens used per problem for each configuration, illustrating the trade-offs between performance and computational cost as the branch limit increases.

