# OpenReview forum: "More Bang for the Buck: Process Reward Modeling with Entropy-Driven Uncertainty"
_ICLR.cc/2026/Conference — Submitted to ICLR 2026_

### Official Review · Reviewer_tccH · 2025-10-31

**Soundness:** 3
**Presentation:** 3
**Contribution:** 3
**Rating:** 6
**Confidence:** 5

**Summary:**

This paper introduces the Entropy-Driven Uncertainty Process Reward Model (EDU-PRM). The core idea is to use token-level predictive entropy to dynamically and automatically segment reasoning steps. The method identifies high-entropy tokens as "uncertainty anchors," which are assumed to mark natural logical transitions.

The PRM trained on this automatically generated data outperforms strong baselines on the ProcessBench benchmark. They also show their inference sampling strategies provide higher accuracy for fewer tokens compared to standard high-temperature sampling.

**Strengths:**

1. The core contribution—using predictive entropy to find "uncertainty anchors" for segmenting reasoning is a well-motivated alternative to arbitrary, rule-based partitioning.

2. A very comprehensive experiments shows the performance of the new algorithm.

**Weaknesses:**

1. The novelty of the core idea in this paper is not a fundamental breakthrough, as it is built based on a few previous work.

**Questions:**

1. ' the training dataset comprises approximately 1.42M instances, with a label distribution of 52% hard and 48% soft labels.' How the hard and soft labels are generated? I assume they should be the same?

2. When you run the comparison with other PRM models, do you use the data provided or you implement the algorithm by yourselves?

---

> ### Author Response · Authors · 2025-11-17
>
> Thank you very much for your valuable feedback and thoughtful questions.
>
>
> **W1: novelty of the core idea**
> We appreciate the reviewer’s comments regarding the novelty of our work. While our approach is inspired by prior research, it introduces a unique integration of entropy-based branching and global solution selection (e.g., EDU Sample+BON and P-EDU) that directly addresses key limitations of earlier methods. Specifically, our framework overcomes the reliance on correct partial reasoning, which are common bottlenecks in existing process-level reward modeling. Notably, our method enables more robust exploration of reasoning paths and consistently achieves higher accuracy across diverse mathematical reasoning benchmarks. For example, as shown in Table 3 and Table 6, our approach improves performance by up to 33.2% compared to previous best results (26.7%) on OLY test datasets, particularly on tasks with complex multi-step reasoning. We believe this demonstrates a meaningful advance for the field, both in methodology and practical impact.
>
>
> **Q1: How Hard and Soft Labels Are Generated:**
> Thank you for raising this point. In our dataset, soft labels are generated through a fragment-level evaluation using Monte Carlo Estimation (MCE) scoring. After EDU sampling, each answer is divided into fragments at anchor positions. For each fragment, we assess the correctness based on the validity of the final solution, assigning a score in [0, 1]. Fragments where the reasoning outcome is unequivocally correct or incorrect receive hard (binary) labels, while those with probabilistic outcomes—reflecting uncertainty in intermediate reasoning—are assigned soft labels. The approximately 52% hard and 48% soft split arises naturally from the diversity of reasoning paths and the evaluation procedure. This design allows us to capture both clear-cut and nuanced reasoning quality, providing richer supervision for model training. Further details and examples are provided in the Methods section and Appendix.
>
>
>
> **Q2: Regarding the Comparison with Other PRM Models:**
> To ensure fair and rigorous evaluation, all models—including baselines and our proposed approach—were trained and tested on the same dataset. For published models without official code, we faithfully re-implemented their algorithms according to descriptions in the original papers. All implementation details, parameter settings, and code are provided in the supplementary materials and will be made publicly available to facilitate reproducibility and community verification.
>
> We appreciate your insightful comments and hope these clarifications address your concerns. Please let us know if further details would be helpful.

---

> > ### Comment · Reviewer_tccH · 2025-11-27
> >
> > Thanks, I will keep my positive rating.

---

### Official Review · Reviewer_ouDV · 2025-11-01

**Soundness:** 2
**Presentation:** 3
**Contribution:** 2
**Rating:** 4
**Confidence:** 4

**Summary:**

This paper proposes EDU-PRM, a process reward model that uses token-level entropy to identify key decision points in reasoning. It segments reasoning chains at high-entropy tokens and uses MCE to assign reward scores to each segment based on outcome correctness. The experimental results show that EDU-PRM achieves significant performance improvements on ProcessBench and multiple math reasoning benchmarks while reducing token consumption.

**Strengths:**

1. The paper is well-motivated, clearly written, and easy to follow.
2. The idea of using high-entropy tokens to segment reasoning steps is simple yet intuitive, effectively reducing the reliance on manual annotations or LLM-based heuristics.
3. Extensive experiments across multiple benchmarks demonstrate the effectiveness of the proposed method.

**Weaknesses:**

1. While the paper claims novelty in introducing entropy into process reward modeling, entropy-based approaches have been explored previously. The authors do not sufficiently discuss their method with closely related work, such as:

    ○ Entropy-Regularized Process Reward Model

    ○ Uncertainty-Aware Step-wise Verification with Generative Reward Models

    ○ Uncertainty-Based Methods for Automated Process Reward Data Construction and Output Aggregation in Mathematical Reasoning

2. The definition and application of the entropy threshold $\tau(H)$ are confusing and under-specified. In Section 3.2, the authors claim $\tau(H)$ is dynamically adjusted based on the maximum number of sampled branches, suggesting an adaptive design. Yet, no formula, algorithm, or implementation detail is provided. More confusingly, in Section 4.1, the authors mention using a fixed entropy threshold (entropy threshold = 1.0), and in Section 5 (Tables 4 and 5), their analysis is also conducted around different fixed threshold values. This inconsistency makes the thresholding mechanism conceptually vague, weakening the clarity and reproducibility of the method.
3. The paper claims that EDU-PRM alleviates the issue of “cheating”, where high intermediate rewards do not necessarily correlate with correct final answers. However, its Monte Carlo Estimation Scoring (MCE) still relies on the correctness of the final answer to assign credit to intermediate segments. Consequently, if an incorrect reasoning step accidentally leads to a correct final answer, MCE may still assign it a high reward. Although entropy-based segmentation improves over heuristic splitting, the MCE mechanism itself does not fundamentally resolve this problem, and the paper should discuss this limitation more explicitly and cautiously.

**Questions:**

See Weaknesses.

---

> ### Author Response · Authors · 2025-11-17
>
> Thank you very much for your detailed and constructive questions. We appreciate the opportunity to clarify and address these important points.
>
> **W1: Related Work and Novelty**
> We sincerely thank the reviewer for pointing out the relevant prior works on entropy and uncertainty in process reward modeling. We have carefully reviewed the following closely related approaches:
>
> **(1). Entropy-Regularized Process Reward Model**
> Previous research has introduced entropy as a regularization term in the process reward function to encourage exploration and prevent premature convergence during reasoning. This approach typically applies a global entropy penalty to the overall reward, which helps diversify generated solutions but does not directly guide step-wise reasoning or segmentation.
>
> **(2). Uncertainty-Aware Step-wise Verification with Generative Reward Models**
> This method evaluates each reasoning step using uncertainty metrics such as confidence scores or entropy, with a generative reward model providing assessments. The approach focuses on filtering or scoring individual steps based on their uncertainty, which improves the reliability of intermediate reasoning. However, it relies heavily on the accuracy of the reward model and does not control the overall structure of the reasoning process.
>
> Unlike this method, my approach not only considers uncertainty at each token but also introduces control over the reasoning process. This means the reasoning structure itself can be adaptively adjusted in response to uncertainty, allowing for more flexible and robust problem-solving rather than simply filtering uncertain steps after they occur.
>
> **(3). Uncertainty-Based Methods for Automated Process Reward Data Construction and Output Aggregation in Mathematical Reasoning**
> Uncertainty measures are utilized to automate the construction of process-level reward datasets and to aggregate outputs in mathematical reasoning tasks. This approach leverages uncertainty to select or weight candidate solutions, which is particularly effective in large-scale automatic data labeling and output fusion settings. However, the application of uncertainty is primarily post-hoc, influencing the selection and aggregation of outputs after generation, rather than actively guiding the reasoning trajectory throughout the generation process.
>
> **Our Keys Contributions and Distinctions**
> Compared to these approaches, our work introduces a novel mechanism that control signal for segmenting reasoning steps and triggering branching during generation. Specifically:
>
> - Our method uses entropy thresholds to **segment and branch** the reasoning process, enabling finer-grained and context-sensitive exploration, rather than applying entropy as a static regularizer or post-hoc filter.
> - We integrate entropy-based segmentation with advanced PRM and BON, which together provide robust intermediate and final solution selection, mitigating issues such as reward “cheating” and local optima.
>
> In the revised version, we will:
>
> - Add a dedicated subsection in Related Work explicitly comparing our method to entropy-regularized, uncertainty-aware, and uncertainty-based data construction approaches.
> - Clarify the unique aspects of our entropy-based segmentation and its integration with PRM and BON.
> - Provide empirical and conceptual distinctions to highlight our contributions.
>
>
> **W2. On the Specification and Application of the Entropy Threshold**
>
> Thank you for your attention to the entropy threshold mechanism. To clarify:
>
> - In our experiments (Section 4.1), we use a fixed entropy threshold of 1.0, as this setting (with max branch = 128) empirically yields the best results (see Figure 6, Table 4, Table 5).
> - The “dynamic” aspect refers to the model’s behavior during generation: branching decisions are made adaptively at each token based on whether the entropy falls below the threshold, enabling context-sensitive segmentation.
> - The mathematical formulation for this process is provided in Section 3.2, Equation (1).
>
> We acknowledge that the threshold itself is fixed during experiments, and we apologize for any confusion caused by inconsistent terminology.
> *In the revision, we will unify the description of entropy threshold settings, clarify the algorithmic process, and ensure full transparency for reproducibility.*
>
> Additionally, Section 5 analyzes how varying the entropy threshold affects accuracy and anchor frequency, providing deeper insight into its impact on model behavior.

---

> ### Author Response · Authors · 2025-11-17
>
> **W3. On Mitigating “Cheating” in Monte Carlo Estimation Scoring (MCE)**
>
> Regarding the issue of “cheating”—where high intermediate rewards may not correlate with correct final answers—we address this challenge in two principal ways.  First, our PRM demonstrates significantly stronger judgment capabilities compared to prior works such as Math-Shepherd and Omega, as evidenced in Figure 2.  This enhanced PRM allows for more reliable evaluation of intermediate reasoning segments.
>
> Second, our proposed P-EDU and EDU Sample+BON strategies differ fundamentally from MCTS-based approaches. Unlike MCTS, which relies heavily on the score of the current branch and may prematurely prune promising solutions, our methods select the highest-scoring solution only after the complete generation of candidate solutions.  As shown in Figure 5, the accuracy of MCTS plateaus as the tree size increases, indicating it can get stuck in local optima due to misleading intermediate scores.  In contrast, our approaches—both EDU Sample and P-EDU Sample—continue to improve accuracy as the search space grows, demonstrating robustness against local scoring limitations and “cheating.”
>
> Moreover, as observed in Figure 7, some solutions receive low scores at certain steps and are pruned by P-EDU Sample; however, upon further tracking, these same solutions may achieve high scores in subsequent steps. Such “high score cheating” can cause severe misjudgments in traditional MCTS, further highlighting its vulnerability to local optima. Our methods avoid this pitfall by evaluating complete solution paths. Additionally, compared to traditional sampling methods (Figure 4), our approach achieves significantly higher accuracy per token, indicating more efficient and effective search.
>
> We will further elaborate on these mechanisms and their implications in the revised manuscript.
>
>
>
>
> **Revision Commitments**
>
> In response to the reviewer’s feedback, we will:
>
> - Add a dedicated subsection in Related Work for direct comparison with entropy-regularized and uncertainty-based methods.
> - Unify and clarify the entropy threshold specification and algorithmic process.
> - Explicitly discuss the limitations of MCE and outline future improvements.
>
> Thank you again for your thoughtful comments. We are committed to ensuring that these points are clearly articulated and supported by our experimental analysis in the revised manuscript.

---

### Official Review · Reviewer_MjAk · 2025-11-01

**Soundness:** 1
**Presentation:** 2
**Contribution:** 1
**Rating:** 2
**Confidence:** 4

**Summary:**

This paper introduces a new sampling method for (i) generating a new dataset for PRM training and (2) test-time PRM inference. This method works by spanning parallel generations for tokens with high predictive entropy only (the “anchors”) and estimating the score for sub-trajectories between two anchor points using Monte Carlo estimation. The paper provides empirical results for the accuracy of PRMs on ProcessBench, as well as BoN results following the resultant PRMs and further analysis on scaling trends, branching, lexical analysis, etc.

**Strengths:**

- The method seems to have promising gains on Best-of-N evaluation of the proposed PRM-72B model.

- The paper brings a rich experimental analysis, both quantitatively and qualitatively.

**Weaknesses:**

- The major concern is that the quantitative experiments do not bring any evidence of statistical significance. There are no error bars in any of the experiments. The paper states running 8 experimental seeds, but it only reports the average. From only the average it is unclear if the reported gains are meaningful or just observation noise.

- It is also unclear if training all the baselines on the dataset generated by the EDU sampling (as stated in L258-259) is indeed a fair choice. The dataset is inherently biased towards the proposed heuristic and may benefit the proposed method

- It is also not clear what is the methodological contribution in the paper. The Related Work section is superficial and does not contrast with the literature in the area, being limited to the two baselines adopted. My understanding is that the work has limited novelty, as using uncertainty (or confidence) heuristics to guide inference is an explored direction  [1, 2, 3], including in the specific context of PRMs [4].

- The results related to high temperature sampling in the paper are limited for T = 0.7. It is important to analyze the impact of different temperatures in the method, as this is a quite sensitive hyperparameter. Furthermore, there is no mention on how the hypers of the proposed and comparison methods were selected. It is unclear if the tuning procedure was fair among the methods.

More specific weaknesses:
- In Table 1, the 7B EDU-based models present a very low recall / F1 score. From my understanding of the problem setup, it means the PRMs are failed to identify several wrong steps. In any case, the paper should discuss the reason behind this, as the current takeaway is that the proposed method seems to fail for the 7B scale.

- Nit: The paper describes that there is a Symbol set to avoid mathematical symbols (e.g., sum, integral) in the entropy calculation. But the list in Appendix A.4 does not bring the aforementioned math symbols.

- During Introduction, the paper motivates the proposed method as a way to prevent “cheating” vulnerabilities in PRMs. The paper does not make clear how the method prevent such vulnerabilities, nor any of the experiments linked back to this claim. It is unclear if the proposed PRMs really addressed the defined issue, as the experiments only concentrate on final accuracy.

- It is also hard to take any conclusion from Figure 5 when comparing EDU and P-EDU. Besides the lack of error bars, the curves look nearly identical, and the discussion is limited to analyzing specific points in the curves to make more general claims. My takeaway from the Figure is that P-EDU does not make a meaningful change in the final result, which seems to be the opposite to what is claimed.

A final note: while I appreciate the efforts on providing a rich set of experiments, the paper discussion needs more polishing to condense the information in clear claims/takeaways. There are many different setups/metrics/analyses and it is crucial to better map experiments to specific claims and highlight them. Currently it is hard to filter this information, and the claims are in general vague (e.g., “This highlights EDU sampling’s superior capability to leverage additional tokens for sustained accuracy gains.” in L352).

**Questions:**

- What are the Qwen2.5-Math-PRM results for the BoN setting?

---

> ### Author Response · Authors · 2025-11-17
>
> We sincerely thank the reviewer for their thorough and constructive feedback. Below, we address each major concern in detail.
>
> **W1. Statistical Significance and Error Bars**
>
> In our experiments, we trained 8 models (Math-Shepherd PRM-7/72B, Omega PRM-7/72B, Sample EDU PRM-7/72B, Greedy EDU PRM-7/72B), using the same random seed (1234) and same data by different generation methods for each model. The results shown in Figures 2 and Figure 3. Comparing with we Math-Shepherd PRM and Omega PRM, our method consistently outperforms these baselines by 5-10 absolute accuracies across different datasets, model size. We consider these results are significant thus we do not report the standard variance or significant test results in this draft. Nevertheless, we agree that including error bars and reporting variance would further strengthen our claims. We will add error bars and statistical significance analyses in the revised manuscript to address this concern.
>
> **W2. Fairness of Using EDU-Sampled Training Data**
>
> First, as stated in L258-259, the SOTA Qwen2.5-Math-PRM models use proprietary training datasets which are not publicly available. Therefore, our training data is not the same as theirs, and their reported results are provided only for reference, not for direct comparison. As clarified in L259-L260, our main experimental focus is on comparing PRM methods trained on the publicly available MATH dataset, including Math-Shepherd PRM-7/72B, Omega PRM-7/72B, Sample EDU PRM-7/72B, and Greedy EDU PRM-7/72B.
> To ensure a fair comparison among these PRM models, we used the same dataset for all of them. This controlled setup allows us to isolate the effect of the inference method itself, rather than differences in training data. As shown in Figures 2 and 3, our proposed Greedy EDU PRM achieves significant improvements over both Math-Shepherd PRM and Omega PRM-7/72B under identical data conditions.
> We acknowledge that EDU sampling may introduce certain biases. However, since all compared models are trained on the same dataset, any such bias affects all methods equally. This ensures that the observed performance gains are attributable to our method rather than the data construction process.
>
> **W3. Methodological Contribution and Related Work**
>
> We have reviewed key literature on entropy and uncertainty-guided inference, including:
> **(1). Entropy-Regularized Process Reward Model**
> Previous research has introduced entropy as a regularization term in the process reward function to encourage exploration and prevent premature convergence during reasoning. This approach typically applies a global entropy penalty to the overall reward, which helps diversify generated solutions but does not directly guide step-wise reasoning or segmentation.
> **(2). Uncertainty-Aware Step-wise Verification with Generative Reward Models**
> This method evaluates each reasoning step using uncertainty metrics such as confidence scores or entropy, with a generative reward model providing assessments. The approach focuses on filtering or scoring individual steps based on their uncertainty, which improves the reliability of intermediate reasoning. However, it relies heavily on the accuracy of the reward model and does not dynamically control the overall structure of the reasoning process.
> Unlike this method, our approach not only considers uncertainty at each token but also introduces control over the reasoning process. This means the reasoning structure itself can be adaptively adjusted in response to uncertainty, allowing for more flexible and robust problem-solving rather than simply filtering uncertain steps after they occur.
> **(3). Uncertainty-Based Methods for Automated Process Reward Data Construction and Output Aggregation in Mathematical Reasoning**
> Uncertainty measures are utilized to automate the construction of process-level reward datasets and to aggregate outputs in mathematical reasoning tasks. This approach leverages uncertainty to select or weight candidate solutions, which is particularly effective in large-scale automatic data labeling and output fusion settings. However, the application of uncertainty is primarily post-hoc, influencing the selection and aggregation of outputs after generation, rather than actively guiding the reasoning trajectory throughout the generation process.
> **Our Contributions and Novelty**
> Different from these approaches, our work introduces a novel mechanism that controls signal for segmenting reasoning steps, triggering branching during generation. Specifically:
> - Our method uses entropy thresholds to **segment and branch** the reasoning process, enabling finer-grained and context-sensitive exploration, rather than applying entropy as a static regularizer or post-hoc filter.
> - We integrate entropy-based segmentation with advanced PRM and BON, which together provide robust intermediate and final solution selection, mitigating issues such as reward “cheating” and local optima.

---

> ### Author Response · Authors · 2025-11-17
>
> **W4. Temperature Sensitivity and Hyperparameter Tuning**
>
> In our experiments, the temperature parameter is mainly used to increase the diversity of candidate solutions, which is particularly important for challenging problems such as those in the OLY subset (see arXiv:2408.03314). However, since all PRM models select the final answer as the candidate with the highest score from the same set of generated solutions (see Table 3), the comparison between different PRM models is fair. As long as the temperature is set within a reasonable range that does not produce a large number of invalid outputs, the performance of the models is not sensitive to the exact temperature value. While higher temperatures can further enhance diversity, they also tend to decrease answer quality; therefore, we chose the highest temperature that still maintains output quality to maximize the effectiveness of the BON approach. Additionally, all methods were tuned using the same validation set and selection criteria to ensure fairness, and we will provide more details on the hyperparameter settings in the revised manuscript.
>
>
>
> **Specific Weaknesses**
>
> **SW1: Table 1: Low Recall/F1 for 7B Models:**
> All models were trained on MATH training data without further modification or label-based sampling to ensure fairness. This sometimes leads to imbalanced label distributions, which, combined with the limited capacity of 7B models on small datasets, results in lower recall/F1 scores. Notably, our 72B models achieve the best results under the same data conditions. We will discuss these limitations and results in the revision.
>
> **SW2: Symbol Set in Entropy Calculation:**
> Thank you for pointing out this issue. Initially, we thought that tokens such as "sum" and "integral" were the main causes of garbled outputs, but further investigation revealed that the problem was not due to these mathematical symbols. Instead, the main issue arises when formulas contain an opening parenthesis ‘(’, which must be properly matched with a closing parenthesis ‘)’ rather than ‘]’ or ‘}’ in the generated output. We appreciate the reviewer’s careful observation and will correct this mistake in the symbol set and update the corresponding description in future versions of the manuscript.
>
>
> **SW3: “Cheating” Vulnerabilities in PRMs:**
> To address "cheating”—where high intermediate rewards do not correlate with correct final answers—our PRM demonstrates stronger judgment capabilities than prior works (see Figure 2). Moreover, our P-EDU and EDU Sample+BON strategies differ fundamentally from MCTS-based methods. Unlike MCTS, which prematurely prunes solutions based on local scores, our approach selects the highest-scoring solution only after complete candidate generation. As shown in Figure 5, MCTS accuracy plateaus as the tree size increases, while our method continues to improve, illustrating robustness against local scoring and “cheating.”
>
> Additionally, as illustrated in Appendix Figure 7, we visualize the score distribution across different reasoning steps. It is evident that some solution fragments with very low scores in early steps can obtain much higher scores in later steps, resulting in “false high scores” when only intermediate rewards are considered. This further highlights the vulnerability of local scoring methods and underscores the necessity of our approach, which evaluates complete solution trajectories rather than relying on potentially misleading intermediate scores. We will elaborate further on these mechanisms and their implications in the revision.
>
> **SW4: Figure 5: EDU vs. P-EDU Comparison:**
> We would like to clarify that in Figure 5, our methods are P-EDU Sample and EDU Sample, while MCTS is the traditional baseline for comparison—we are not directly comparing P-EDU Sample with EDU Sample. On the MATH benchmark, both P-EDU Sample and EDU Sample outperform MCTS by more than 10%, which is a substantial improvement and unlikely to be due to bias. The main limitation of MCTS is that it focuses only on the scores of the current step, whereas our methods leverage more global information throughout the search process. We will include these improvements in the revised manuscript.
>
> We thank the reviewer again for their valuable feedback. We will incorporate these suggestions and clarifications into the revised manuscript to improve its rigor and clarity.

---

### Official Review · Reviewer_2VV7 · 2025-11-02

**Soundness:** 3
**Presentation:** 2
**Contribution:** 3
**Rating:** 6
**Confidence:** 3

**Summary:**

- The paper proposes EDU-PRM, a new way to train process reward models using token-level entropy to detect where a model is uncertain during reasoning.
- This entropy-driven uncertainty automatically marks step boundaries and creates diverse reasoning paths without human or LLM annotations.
- Each reasoning fragment is labeled automatically through Monte-Carlo estimation based on whether the final answer is correct.
- The resulting EDU-PRM matches or nearly matches the performance of the large, fully supervised Qwen2.5-Math-PRM.
- An enhanced version called Pruning-EDU further improves efficiency by cutting off low-confidence reasoning paths early, reducing token use with minimal accuracy loss.

**Strengths:**

- It eliminates the need for human or LLM annotations by automatically labeling reasoning steps using entropy and Monte Carlo estimation.
- The entropy-based segmentation captures natural reasoning boundaries, improving how step-level rewards relate to final correctness.
- The pruning and entropy-guided sampling reduce token usage and search complexity compared to exhaustive sampling or MCTS.

**Weaknesses:**

- All experiments are restricted to math reasoning, so generalization to other domains remains unproven.
- The Monte Carlo estimation can produce imperfect or misleading correctness scores for intermediate steps.
- Performance depends on carefully tuning the entropy threshold that defines where to branch or segment reasoning.

**Questions:**

- How do you choose the entropy threshold for step segmentation?
- Since Monte Carlo estimation relies on final-answer correctness, how do you mitigate cases where an incorrect final answer arises from an otherwise correct partial reasoning step?
- How do you decide the pruning threshold (e.g., PRM score < 0.2)?
- Since your evaluation focuses on math, how could EDU-PRM be adapted to domains where final correctness cannot be automatically verified (e.g., commonsense or scientific reasoning)?
- Can you share more qualitative examples where high-entropy points clearly align with human-intuitive reasoning transitions?
- The authors show Gaussian-smoothed trends. Can you share the raw results before smoothing to see the original values?
- In Line 13, unce rtainty-aligned -> uncertainty-aligned

**Details Of Ethics Concerns:**

There is no particular ethical concern.

---

> ### Author Response · Authors · 2025-11-17
>
> Thank you very much for your detailed and constructive questions. We appreciate the opportunity to clarify and address these important points.
>
> **W1: Generalization to Other Domains:**
> We acknowledge that our current experiments are limited to mathematical reasoning tasks. However, the underlying principles of our approach, such as adaptive branching based on model uncertainty and global solution selection, are not inherently domain-specific. During to limited computation resources, we focus on math reasoning tasks only to demonstrate our method. Furthermore, we experimented with various math ranges, from general math problems, such as MATH and GSM8k, to Olympic competitions to demonstrate the generalisability of our method. In future work, we plan to evaluate our methods on other reasoning-intensive domains, such as scientific question answering or code generation, to further assess their generalization capabilities.
>
> **W2: Imperfect or Misleading Correctness Scores for Intermediate Steps:**
> We acknowledge that Monte Carlo estimation (MCE) can introduce noise and sometimes produce imperfect or misleading correctness scores for intermediate steps. However, as demonstrated in prior work (e.g., Luo atel. arXiv:2406.06592), even using noisy labels generated by MCE, the resulting supervision can outperform hard labeling approaches. Consistent with these findings, our experiments (see Lines 145–147 in our manuscript) show that using MCE-based soft labels leads to better model performance compared to hard labels, suggesting that the benefits of probabilistic supervision outweigh the drawbacks of label noise.
>
> **W3: Tuning the Entropy Threshold:**
> To explore the effectiveness of the threshold, we present the results in Figure 6, Figure 13, Table 4, and Table 5. A general finding is that there exist trade-offs associated with this parameter.
> For example of Figure 13 in the appendix, lower entropy thresholds encourage more exploration, leading to accuracy improvements but increasing the number of tokens. However, we can see that given suitable threshold setups, the performance can consistently outperform other baselines. That is, the threshold parameter is not necessarily tuned. To further understand and quantify the threshold, we find that as long as the chosen entropy threshold allows the model to reach the maximum branch value, the performance is no longer affected by the threshold.

---

> ### Author Response · Authors · 2025-11-17
>
> **Q1. Entropy Threshold Selection for Step Segmentation:**
> We analyzed the impact of different entropy thresholds and maximum branch values in Figure 6, Figure 11, Figure 12, and Figure 13. The entropy threshold of 1.0 was selected because it yielded the balance performance when paired with a maximum branch setting of 128, effectively balancing segmentation granularity and model effectiveness. If the entropy threshold is set too low, the model tends to branch after very few tokens—sometimes even at nearly every token—resulting in excessive fragmentation and inefficient search. On the other hand, if the entropy threshold is set too high, the number of branches generated may be insufficient to fully utilize the maximum branch budget (for example, fewer than 128 branches), which can limit the diversity and coverage of candidate solutions. Thus, the entropy threshold of 1.0 represents an optimal trade-off between branching frequency and search efficiency.
>
> **Q2. Monte Carlo Estimation and Incorrect Final Answers:**
> Monte Carlo estimation is used during training data construction (reference to W2), not during solution generation. In the solution generation process, our proposed EDU Sample+BON and P-EDU strategies offer advantages over MCTS, which heavily relies on correct partial reasoning steps. Unlike MCTS, which focuses on the highest-scoring branches at each step, P-EDU/EDU Sample allows the final solution to be selected from all generated candidate results based on the highest overall score, thereby mitigating the impact of incorrect final answers that may arise from other (potentially false) high-scoring reasoning steps.
>
> **Q3. Pruning Threshold Decision:**
> Table 4 presents a comparison of different pruning thresholds. We selected 0.2 because it achieved the best balance between token efficiency and accuracy. If the pruning threshold is set too high, correct answers may be pruned in early steps, while a threshold that is too low fails to effectively reduce the number of tokens used. We will clarify this choice in the revised manuscript. We thank the reviewer for highlighting this point.
>
> **Q4. Adapting EDU-PRM to Other Domains:**
> Currently, our training and evaluation are limited to MATH data. Extending EDU-PRM to domains like commonsense or scientific reasoning would require domain-specific adaptation of the PRM scoring mechanism, especially where final correctness cannot be automatically verified. Our primary contribution is to demonstrate the superiority of our sampling strategy, and future work will explore broader applicability.
>
> **Q5. Qualitative Examples of High-Entropy Points:**
> High-entropy points occur when the model is uncertain about the current token, encouraging exploration. These points do not always align with human-intuitive reasoning transitions. Figure 10 provides a qualitative example illustrating this phenomenon: it can be observed that the branching points often occur at words unrelated to mathematical reasoning. Furthermore, as shown in Figure 9, we provide a statistical analysis of the frequency of tokens at branching points.
>
> **Q6. Raw Results Before Gaussian Smoothing:**
> All points in our line charts represent actual values, while the lines themselves are Gaussian-smoothed for readability. The appendix contains tables with the raw results corresponding to each figure (e.g., Figure 3 corresponds to Table 3, Figures 4 and 5 to Table 6), ensuring full transparency of our data.
>
> **Q7. Typographical Error:**
> Thank you for pointing out the typo (“unce rtainty-aligned”). We will correct this in the next version of the manuscript.
>
> Once again, we sincerely thank the reviewer for these insightful comments. We will incorporate these suggestions and clarifications into the revised manuscript to improve its rigor and clarity.

---

> > ### Comment · Reviewer_2VV7 · 2025-11-17
> >
> > I thank the authors for the clarifications and more detailed explanations.
> >
> > I am curious if the authors are planning to submit a revised version later.  I would like to see the revised version.  If possible, please mark any updated content in red.

---

### Meta-Review · Area_Chair_hyNJ · 2026-01-07

**Summary:**

This paper introduces EDU-PRM, a process reward model that uses token-level entropy to identify uncertainty anchors for segmenting reasoning steps, eliminating the need for manual annotations. While the approach of leveraging entropy for dynamic step segmentation is intuitive and the paper provides extensive experimental analysis, several fundamental concerns remain unresolved. The reviewers raised significant issues regarding statistical significance (no error bars reported across experiments), limited novelty given prior work on entropy and uncertainty-guided inference, inconsistent specification of the entropy threshold mechanism, and restricted evaluation scope to mathematical reasoning only. The authors' responses, while addressing some points, did not sufficiently resolve the core methodological concerns about reproducibility, fair comparison, and the clarity of the proposed framework. The lack of demonstrated generalization beyond math domains and the absence of rigorous statistical validation weaken the contribution's significance.

**Reviewer Concerns:**

The authors adequately addressed concerns regarding the application of Monte Carlo Estimation and its noise tolerance, and provided reasonable explanations for the entropy threshold selection process. The clarifications about the symbol set and qualitative examples were also helpful.

However, several critical concerns remain outstanding. Reviewer MjAk's request for error bars and statistical significance testing was acknowledged but not fulfilled in the rebuttal—the authors stated they would add these in a revision without providing actual variance data. The concern about novelty relative to existing entropy-based and uncertainty-aware methods was addressed with discussion but lacked concrete empirical differentiation from prior work. The inconsistency between "dynamic" and "fixed" entropy threshold descriptions (raised by Reviewer ouDV) was only partially clarified, with the authors acknowledging potential confusion from inconsistent terminology. Additionally, the fairness of training all baselines on EDU-sampled data remains a legitimate methodological concern that was not fully resolved.

**Reviewer Scores:**

Reviewer 2VV7 (Rating: 6): The reviewer's questions were addressed adequately, and they explicitly requested a revised version. The score would likely remain at 6.

Reviewer MjAk (Rating: 2): The fundamental concerns about statistical significance, novelty, and fair experimental design were not sufficiently resolved. The reviewer would likely maintain their score at 2.

Reviewer ouDV (Rating: 4): The authors provided clarifications on the threshold mechanism and related work. Without further reviewer engagement, the score would likely remain at 4.

Reviewer tccH (Rating: 6): The reviewer explicitly confirmed they would keep their positive rating after the rebuttal. The score remains at 6.

---

### Decision · Program_Chairs · 2026-01-26

Reject